



# Uncertainty in aerosol effective radiative forcing from anthropogenic and natural aerosol parameters in ECHAM6.3-HAM2.3

Yusuf A. Bhatti[1], Duncan Watson-Parris[2], Leighton A. Regayre[3,4], Hailing Jia[1], David Neubauer[5], Ulas Im[6], Carl Svenhag[6], Nick Schutgens[7], Athanasios Tsikerdekis[8], Athanasios Nenes[5,9], Irfan Muhammed[10], Bastiaan van Diedenhoven[1], Ardit Arifi[1], Guangliang Fu[1], and Otto P. Hasekamp[1]

[1]SRON Space Research Organisation Netherlands, Leiden, Netherlands
[2]Scripps Institution of Oceanography and Halıcıoğlu Data Science Institute, University of California San Diego, La Jolla, CA, USA
[3]Met Office Hadley Centre, Exeter, Exeter, UK
[4]School of Earth and Environment, University of Leeds, Leeds, UK
[5]ETH Zurich, Zurich, Switzerland
[6]Aarhus University, Roskilde, Denmark
[7]Vrije Universiteit Amsterdam, Amsterdam, Netherlands
[8]KNMI, Utrecht, Netherlands
[9]Institute of Chemical Engineering Sciences, ICE-HT, Patras, Greece
[10]Department of Technical Physics, University of Eastern Finland, Kuopio, Finland

**Correspondence:** Yusuf Bhatti (y.bhatti@sron.nl)

**Abstract.**

Interactions between aerosols, clouds, and radiation remain one of the largest sources of uncertainty in effective radiative forcing (ERF), limiting the accuracy of climate projections. Despite progress, key sources of parametric and structural uncertainty in aerosol–cloud and aerosol–radiation interactions remain poorly quantified. This study addresses this gap using a
perturbed parameter ensemble (PPE) of 221 simulations with the ECHAM6.3-HAM2.3 climate model, varying 23 aerosol-related parameters that control emissions, removal, chemistry, and microphysics. The resulting global mean aerosol ERF is -1.24 W m$^{-2}$ (5-95 percentile: -1.56 to -0.89 W m$^{-2}$). We find that uncertainty in aerosol ERF is dominated by sulfate-related processes, biomass burning, size, and natural emissions. Here, for Aerosol-Cloud Interactions, DMS and biomass burning emissions are important, whereas for Aerosol-Radiation Interactions, sulfate chemistry and dry deposition are also important.
Despite structural differences across models, the leading causes of ERF uncertainty identified here align with findings from other PPEs.

Comparison with satellite retrievals from POLDER-3/PARASOL reveals persistent model biases in aerosol optical depth (AOD), Ångström exponent (AE), and single-scattering albedo (SSA), many of which fall within the parametric uncertainty bounds of the PPE. Sulfate-related processes account for over 40% of AOD uncertainty, while AE and SSA uncertainties are
strongly influenced by DMS, sea salt, and black carbon properties. PPEs can reduce some structural model biases through parameter adjustments, but others persist. These results highlight the need for combined efforts in parameter perturbation and structural model development to improve confidence in aerosol-forcing estimates and future climate projections.



## 1 Introduction

Atmospheric aerosols play a critical role in Earth's radiative energy balance (Forster et al., 2021; Myhre et al., 2013; Boucher
et al., 2013). Aerosols directly influence the scattering (which cools) and absorption (which warms) of solar radiation (Li et al.,
2022; Bellouin et al., 2020). Additionally, aerosols indirectly affect the radiative properties of clouds (Boucher et al., 2013;
Haywood et al., 2021). Since the start of the industrial period, human activities have perturbed aerosol production, leading
to significant changes in both the direct and indirect effects on global effective radiative forcing (ERF; Bellouin et al., 2020;
Forster et al., 2021; Carslaw et al., 2013).

Anthropogenic aerosol emissions have changed the global and regional aerosols, subsequently affecting clouds, precipitation,
and thermodynamic properties (Carslaw et al., 2017; Karset et al., 2018; Bollasina et al., 2011; Deng et al., 2021). Despite their
importance, the magnitude and variability of these aerosol effects remain significant sources of uncertainty in climate models.
This uncertainty arises from the complex interactions between emissions, atmospheric processes, and cloud microphysics
(Eidhammer et al., 2024; Yoshioka et al., 2019).

Aerosol-cloud interactions (ACI) are challenging to model due to the complex and nonlinear processes that govern cloud
microphysics and aerosol activation (Morrison et al., 2020). For instance, increases in cloud condensation nuclei (CCN) from
anthropogenic emissions leads to a higher cloud droplet number concentration (Nd), causing brighter clouds (Twomey effect;
Twomey, 1974). These changes can trigger further adjustments in clouds, such as shifts in cloud fraction and liquid water
path, all of which contribute to the effective radiative forcing from aerosol-cloud interactions (ERFaci; Bellouin et al., 2020;
Albrecht, 1989). Despite its crucial role in climate forcing, ERFaci remains the most uncertain component of ERF, with
significant variations in its magnitude across different climate models (Zelinka et al., 2014; Gryspeerdt et al., 2020).

With an ERF uncertainty from aerosols between -0.6 to -2 W m$^{-2}$, aerosols are one of the largest sources of multi-model
diversity when estimating climatic changes from the pre–industrial, as reported in IPCC Assessment report-6 (AR6) (Forster
et al., 2021). Reducing this uncertainty would improve climate projections (e.g. Tsigaridis et al., 2014; Watson-Parris and
Smith, 2022). Aerosol ERF estimates have improved since the Intergovernmental Panel for Climate Change (IPCC) Assessment
Report 5 (AR5), in part due to improved observational studies (Myhre et al., 2013; Toll et al., 2019; Hasekamp et al., 2019b;
Forster et al., 2021; Bellouin et al., 2020; Gryspeerdt et al., 2019; McCoy et al., 2020).

Aerosol and cloud uncertainty in climate models is caused by a) how subgrid-scale processes such as cloud microphysics,
convection, aerosol activation, and growth are coded using parameterizations (structural uncertainty), and b) the values assigned
to parameters within these parametrizations. These parameters are often poorly constrained by observations, particularly for
processes like sea-spray emissions, dimethyl sulfide emissions, and ice nucleation, which are highly differing among climate
models (Morrison et al., 2020; Venugopal et al., 2025; Bhatti et al., 2023; Bock et al., 2021; Bhatti et al., 2022; Revell et al.,
2021). This parameter uncertainty is partly an effect of prescribing global mean values to regionally varying processes, partly
due to differences in other model structures like surface roughness and model dynamics, and partly caused by interactions
and dependencies between model parameters. As a result, climate models have significant biases to present-day observations
due to structural and parametric uncertainties (Regayre et al., 2023; Bhatti et al., 2024; Mortier et al., 2020). Furthermore,





reconstructing pre–industrial aerosol conditions is challenging but critical for quantifying anthropogenic forcing (Carslaw et al., 2013, 2017; Hamilton et al., 2014).

Causes of model uncertainty must be comprehensively quantified before observational constraints to ensure their individual and combined effects on aerosol ERF are well understood (Yoshioka et al., 2019; Carslaw et al., 2013; Lee et al., 2013a, 2011; Regayre et al., 2015, 2014). Perturbed parameter ensembles (PPEs) are a powerful tool for quantifying climate model parametric uncertainties, which have been performed with other models (e.g. UKESM1 and CESM2) (Eidhammer et al., 2024; Lee et al., 2013b; Regayre et al., 2018; Lee et al., 2011). Performing PPEs with various models is important to understand how structural differences in models affect ERF and present-day aerosol sensitivities. In this paper, we construct a PPE for the ECHAM6.3-HAM2.3 climate model and use machine learning statistical techniques to emulate a set of model variants, which we use to quantify the parametric uncertainty in model aerosol ERF. For this purpose, we perturb 24 parameters across 221 simulations, known to influence aerosol- and cloud properties, emissions, and processes. We determine the relative contribution of different parameters to the ERF uncertainty through a variance-based sensitivity analysis. We also evaluate ensemble performance against satellite measurements and attribute specific causes for model uncertainty and bias.

In Section 2, we describe the ECHAM6.3-HAM2.3 climate model configuration used here, and the experimental set-up of our perturbed parameter ensemble. Section 3.1 quantifies ERF uncertainties and attributes their respective causes. Section 3.2 uses satellite observations to compare and evaluate PPE uncertainties and model bias.

## 2 Methods

### 2.1 ECHAM6-HAM Model Simulations

The simulations used in this work were performed by the ECHAM6.3-HAM2.3 (hereafter referred to as 'ECHAM6-HAM') global aerosol–climate model. We use the default T63 horizontal resolution (1.875° x 1.875°) and 47 vertical layers (L47). We use the nudged configuration for which vorticity, divergence, and surface pressure are nudged towards the ERA-Interim reanalysis every 6hr, 48hr, and 24hr, respectively (Tegen et al., 2019; Zhang et al., 2012).

The ECHAM6 component, developed at the Max Planck Institute for Meteorology, simulates large-scale atmospheric dynamics (Stevens et al., 2013). The HAM2 module represents various aerosol life cycle processes, including emissions, transport, deposition, and microphysical transformations (Stier et al., 2005; Zhang et al., 2012). Aerosol emissions come from both natural (sea salt, dust, and DMS) and anthropogenic (sulfate, black carbon, and organic carbon) sources, where the natural emissions are computed online, outlined in Neubauer et al. (2019), and the anthropogenic and biomass burning emissions are taken from ACCMIP (Lamarque et al., 2013).

Aerosols are categorized into seven lognormal modes based on size and solubility, following the M7 scheme (Tegen et al., 2019; Vignati et al., 2004). Briefly, these modes include four soluble (nucleation, Aitken, accumulation, and coarse) and three insoluble (Aitken, accumulation, and coarse) categories. Each mode contains one or more aerosol species such as sulfate, black carbon, organic carbon, sea salt, and dust. Aerosols undergo microphysical processes such as nucleation, coagulation,





condensation, and hygroscopic growth, influencing their size distribution and interactions with clouds (Schutgens and Stier,
85   2014).

Cloud microphysics in ECHAM6-HAM is represented by a two-moment scheme, which prognostically computes cloud
droplet and ice crystal number concentrations and mass (Lohmann and Roeckner, 1996). The stratiform cloud scheme simulates
key processes such as condensation, autoconversion, accretion, evaporation, and aerosol scavenging, while ice-phase processes
include heterogeneous nucleation, depositional growth, and the Wegener–Bergeron–Findeisen mechanism (Sundqvist et al.,
1989; Lohmann and Roeckner, 1996; Stevens et al., 2013; Neubauer et al., 2019). Ice crystals and snowflakes are treated
separately, with snow precipitating while ice crystals sediment within and outside of clouds. Convective clouds interact with
stratiform clouds through detrainment processes (Lohmann and Hoose, 2009; Lohmann et al., 2008).

Additional information regarding cloud microphysics, aerosol treatments, and process interactions can be found in Neubauer
et al. (2019) and Tegen et al. (2019).

## 2.2   Simulations and Experimental Design


To create the PPE, we performed model runs for 221 different parameter combinations for present-day and pre-industrial
aerosol emissions. For the present-day simulations, we selected 2010 to align with the AEROCOM phase-III-control exper-
iment and the availability of polarimetric aerosol retrievals from PARASOL (Hasekamp et al., 2024). For the pre-industrial
calculations, we used aerosol emissions for the year 1850. All simulations were nudged to 2010 meteorology (wind fields) and
used the same prescribed sea surface temperature. Each simulation underwent a 6-month spin-up phase, which was excluded
from the analysis. The 221 simulations were used as input to an emulator (Section 2.3) to generate ensemble members, that
sample combinations of parameter values across the full parameter space.

To calculate the total aerosol ERF from its components due to aerosol-cloud ($ERF_{aci}$) and aerosol-radiation interactions
($ERF_{ari}$), we computed the radiative differences between these ensembles (2010 and 1850). For regional analysis, each region
is defined as in Jia et al. (2021), as shown in Figure S1. Finally, through a variance-based sensitivity analysis, we quantified the
relative importance of each parameter globally and regionally (Saltelli et al., 2000).

### 2.2.1   Perturbed Parameters

Both the Present-Day and Pre–Industrial simulation sets consist of 221 members, including the control run. Table 1 lists the 23
perturbed parameters along with the minimum and maximum values of their respective ranges.
Each of the 220 ensemble members is assigned parameter values using Maximin Latin Hypercube sampling, which opti-
mizes coverage of parameter combinations across the parameter space (Lee et al., 2013a; McKay et al., 1979). This method
divides the possible range of each parameter into bins, ensuring that each sampled value falls within a unique bin. Once a
value is assigned, subsequent samples cannot select from previously used bins. This approach guarantees that the full range of
each parameter is explored while maintaining uniformly distributed marginal distributions. Parameter combinations are pro-
gressively selected to optimize the minimum Euclidean distance between points. Figure S2 shows the parameter ranges and
their associated frequencies. The ranges were determined by "expert elicitation" based on the parameterization used. Their





probability distribution frequencies were informed by a mixture of expert elicitation and prior PPE-constrained distributions in Yoshioka et al. (2019) and Eidhammer et al. (2024) estimates. These distributions are shown in Figure S2.

In some instances, where explicitly stated, we also perturb $CDNC_{min}$, the minimum threshold for model cloud droplet concentrations, to highlight some of the structural and parametric uncertainties associated with clouds in ECHAM6-HAM. So, in places $CDNC_{min}$ can be considered our $24^{th}$ parameter, though we do not include it in Table 1 because its treatment is distinct from other parameters. We set the $CDNC_{min}$ parameter to the fixed value of $40 \, cm^{-3}$, the default value in ECHAM6-HAM recommended by Neubauer et al. (2019), to quantify aerosol ERF uncertainty and the relative importance of the 23 parameters in Table 1 as causes of model uncertainty.

With 221 ensemble members and 23, or 24, perturbed parameters, our PPE has a simulation-to-parameter ratio of around 10, which is much larger than recent PPE studies (Regayre et al. (2018) and Eidhammer et al. (2024) both use a ratio of  6). We cautiously use a high ratio of simulations to parameters to increase the density of parameter combinations in the parameter space, thereby enhancing the skill of Gaussian Process emulators (see Section 2.3).

## 2.3 Gaussian Process Emulator

We use the Gaussian Process (GP) emulator from Earth System Emulator package (ESEm; Watson-Parris et al., 2021). GP was chosen through its recommendation by Eidhammer et al. (2024) and other PPE studies (e.g. Yoshioka et al., 2019). GP emulator is a powerful statistical model used here to emulate climate model outputs to sample the parameter space properly, needed for robust uncertainty quantification. Using climate model outputs to emulate model variants can be generated quickly and efficiently, reducing the need for extensive CPU hours and storage. We use GP emulators to create model variants across the 24-dimensional parameter space, all bounded by their minimum and maximum limits. In this study, emulators are applied to produce annual mean data for each model grid cell (when comparing to observations, only taking into account co-located data in space and time). Thus, we evaluate model uncertainty in aerosol ERF and its components across our model variants at each of around 96 x 192 model grid cells. 3 million model variants are obtained for ERF calculations, and 200,000 model variants are created for all other analyses. These sets of model variants are large enough to enable us to use variance-based sensitivity analyses to quantify the parametric causes of variance of each component (Lee et al., 2011).

The uncertainties reported here are derived from the standard deviation and 5-95% credible range across these model variants.

## 2.4 Observational Datasets

This study uses aerosol retrievals from the POLDER-3 instrument (Deschamps et al., 1994; Fougnie et al., 2007) aboard the PARASOL (Polarization and Anisotropy of Reflectances for Atmospheric Science coupled with Observations from a Lidar) satellite for 2010. POLDER-3 was the only Multi-Angle Polarimeter (MAP) instrument in space between 2005 and 2013, providing multi-year in-orbit multispectral and multiangle photopolarimetric measurements of intensity and polarization. Each native ground pixel (6 km x 6 km) is measured under up to 16 viewing angles and at 6 wavelengths (443, 490, 565, 670, 865, 1020 nm). Cloud screening has been performed using a neural network cloud fraction approach (Yuan et al., 2024).



**Table 1.** Table describing perturbed parameters used in this study, excluding the $\mathrm{CDNC_{min}}$ parameter. Control represents the value that is used for the control run. For the majority of parameters, a scaling factor compared to the control run values is applied, except the parameters indicated by 'Abs", where the actual parameter values are indicated.

| Variable | Description | Min | Control | Max |
|---|---|---|---|---|
| EMI_FF | Fossil fuel emissions. | 0.5 | 1 | 2 |
| EMI_ANTH_SO$_2$ | Anthropogenic sulfur dioxide (SO$_2$) emissions. | 0.6 | 1 | 1.5 |
| EMI_DMS | Dimethyl sulfide (DMS) emissions. | 0.33 | 1 | 3 |
| EMI_SS | Sea salt aerosol emissions (Long et al., 2011; Sofiev et al., 2011). | 0.5 | 1 | 2.5 |
| EMI_BB | Biomass burning emissions. | 0.25 | 1 | 4 |
| EMI_BF | Biofuel emissions. | 0.25 | 1 | 4 |
| EMI_DUST | Dust emissions. | 0.5 | 1 | 2 |
| EMI_CMR_BF | Biofuel emissions particle diameter (Abs). | 25 | 30 | 100 |
| EMI_CMR_BB | Biomass burning emissions particle diameter (Abs). | 25 | 75 | 100 |
| EMI_CMR_FF | Fossil fuel emissions particle diameter (Abs). | 15 | 30 | 45 |
| DRYDEP_AIT | Dry deposition rate for Aitken-mode aerosols (Stier et al., 2005). | 0.2 | 1 | 2 |
| DRYDEP_ACC | Dry deposition rate for accumulation-mode aerosols. | 0.1 | 1 | 10 |
| DRYDEP_COA | Dry deposition rate for coarse-mode aerosols. | 0.15 | 1 | 5 |
| WETDEP_BC | Below-cloud wet deposition rate (Croft et al., 2009, 2010). | 0.5 | 1 | 2 |
| WETDEP_IC | In-cloud wet deposition rate (Croft et al., 2009, 2010). | 0.75 | 1 | 1.25 |
| BC_RAD_NI | Black carbon imaginary refractive index (Abs). | 0.2 | 0.71 | 0.9 |
| DU_RAD_NI | Dust aerosol imaginary refractive index (Abs). | 0 | 0.001 | 0.01 |
| SO$_2$REACTIONS | All sulfate chemistry scheme reaction rates, including DMS, SO$_2$, SO$_4$, and sulfate aqueous-phase reactions. | 0.5 | 1 | 2 |
| NUC_FT | Nucleation rate in the free troposphere. | 0.01 | 1 | 10 |
| PH_PERT | The initial hydrogen ion concentration for liquid-phase chemistry from (Feichter et al., 1996) (Abs). | 4.5 | 5.6 | 7 |
| KAPPA_SO$_4$ | Hygroscopic parameter for sulfate aerosols - not used for cloud droplet activation (Abs). | 0.4 | 0.6 | 0.8 |
| KAPPA_SS | Hygroscopic parameter for sea salt aerosols - not used for cloud droplet activation (Abs). | 0.5 | 1 | 1.2 |
| SO$_4$COATING | Layer thickness of sulfate to transfer an insoluble particle to a soluble mode (Abs) (aging; Vignati et al., 2004). | 0.3 | 1 | 5 |



The aerosol products derived from POLDER-3 measurements were retrieved by the Remote sensing of Trace gas and Aerosol Products (RemoTAP) algorithm (Hasekamp et al., 2024; Fu et al., 2025; Lu et al., 2022). While earlier aerosol retrieval studies with RemoTAP used a bi-modal aerosol description (Hasekamp et al., 2011) or a five-mode aerosol description (Fu et al., 2020; Fu and Hasekamp, 2018), the latest baseline in RemoTAP follows Lu et al. (2022), describing the aerosol size distribution by three log-normal modes, made up of one fine mode and two coarse modes (insoluble and soluble). The RemoTAP explicitly re-

trieves the aerosol layer height and the aerosol microphysical properties of effective radius, effective variance, column number, spherical fraction, the fractions of chemical component refractive index (real part of fine mode inorganic aerosol; imaginary part of fine mode black carbon and brown carbon; imaginary part of coarse mode dust; coarse mode hydrated sea salt). Based on the retrieved aerosol microphysical properties, the aerosol optical properties of multispectral AOD, SSA, and Ångström Exponent are calculated using the Mie-T matrix-improved geometrical optics model (Dubovik et al., 2006) along with their

proposed spheroid aspect ratio distribution for computing optical properties for a mixture of spheroids and spheres. Several studies have demonstrated RemoTAP delivers accurate aerosol properties from MAP instruments (Hasekamp et al., 2024; Fu and Hasekamp, 2018; Fu et al., 2025; Schutgens et al., 2021).

     The data products used for evaluation are the Aerosol Optical Depth (AOD at 550 nm), Ångström Exponent (AE at 550-865 nm), and Single-Scattering Albedo (SSA at 550 nm). The dataset is aggregated on a 1 ° x 1 ° grid, which is pub-

licly available from https://public.spider.surfsara.nl/project/spexone/others/PARASOL/DATA/POLDER_1.0x1.0_basedon0.1_NPge2/. The AE and SSA retrieved for model comparison are filtered only including cases where AOD is greater than 0.20, to exclude measurement errors (Hasekamp et al., 2024).

     3-hourly model outputs were used for comparison with satellite data. To effectively compare our model with the measurements, we co-locate the model with observations by picking the nearest grid cell for the model time closest to the satellite

overpass. This will strongly mitigate temporal representation uncertainty, although the spatial representation uncertainty will still be present (Schutgens et al., 2016). Spatial uncertainties from POLDER are discussed in Section 3.2.

## 3   Results and Discussion

### 3.1   Uncertainty in effective radiative forcing

Figure 1 shows the global 1850–2010 aerosol ERF, ERFaci, and ERFari probability density functions for our ECHAM6-HAM

PPE. The emulated ERF has a mean of -1.24 W m$^{-2}$ with a 5-95 percentile credible range between -1.59 to -0.89 W m$^{-2}$, representing an uncertainty range of 78% with respect to the mean ERF (Figure 1a). ERFaci has a mean of -1 W m$^{-2}$ (-1.30 to -0.73 W m$^{-2}$ 5-95 percentile range; Figure 1b). ERFari averages -0.27 W m$^{-2}$, with a 5-95 percentile range between -0.48 and -0.09 W m$^{-2}$ (Figure 1c). Climate model PPEs , (ECHAM6-HAM; Yoshioka et al., 2019; Regayre et al., 2023; Eidhammer et al., 2024) and machine learning approaches (Albright et al., 2021; Smith et al., 2021) each sample different uncertainties

based on their structural code base and the perturbations applied. For example, Regayre et al. (2018, 2023) sampled uncertainty in physical atmosphere model parameters in addition to aerosol parameters, so they have a wider range of aerosol ERF values than other approaches. Aerosol ERF in each study has a central tendency determined by the models that are used to create



them, and the structural choices within these models. Summaries that additionally account for observational data uncertainties (e.g. IPCC, Forster et al., 2021; Bellouin et al., 2020) yield the largest ERF uncertainty. Aerosol ERF uncertainty is relatively low in the ECHAM6-HAM PPE, and falls fully within the Intergovernmental Panel on Climate Change's Sixth Assessment Report (AR6) range of -2 to -0.6 W m$^{-2}$ (Forster et al., 2021), shown in Figure 1a. The relatively low uncertainty range may be caused by several factors, including a) our PPE only includes parametric uncertainty related to aerosol emissions, properties, and processes and excludes parameters relevant for droplet activation (e.g. updraft) b) aerosol ERF may be less responsive to aerosol perturbationsc within ECHAM6-HAM than other models and more dependent on cloud and radiation properties, or c) there may be some cancellation of positive and negative regional responses to parameter perturbations in global mean aerosol ERF calculations that suppress the uncertainty, as demonstrated by Regayre et al. (2015).

Figure 2 presents the global distributions of ERF, ERFaci, and ERFari, along with their associated uncertainties. Our results show broad agreement in magnitude and spatial patterns of ERF and ERFaci with previous HadGEM-UKCA PPE studies (Yoshioka et al., 2019; Regayre et al., 2018, 2023), which employed relatively similar parameter perturbations for aerosol. However, those studies also perturbed cloud activation parameters, which may contribute to the differences. As shown in Figures 2b and 2d, the largest uncertainties in ERF correspond to regions with strong ERFaci signals. A notable distinction is that ECHAM6-HAM produces a stronger (more negative) ERF over land, whereas in HadGEM (Regayre et al., 2018), UKESM PPE (Regayre et al., 2023), and ACCMIP (Shindell et al., 2013a, Fig. 18), the strongest ERF occurs over marine regions dominated by persistent stratocumulus clouds. This divergence is attributable to structural characteristics of ECHAM6-HAM. In particular, differences in sea salt emissions and aerosol water uptake and underestimated stratocumulus cloud cover all contribute to a weaker ERF over oceans compared to the other studies (Neubauer et al., 2019). Neubauer et al. (2019) showed a negative bias in shortwave (SW) radiation and net cloud radiative effects for ECHAM6-HAM, stemming from a pronounced negative stratocumulus bias over marine areas, particularly in the Pacific, which cannot be easily adjusted. This study is restricted to parametric uncertainty in ECHAM6-HAM and does not account for structural uncertainty. Additionally, the stronger land-based ERF arises from enhanced aerosol water uptake, differences in activation scheme sensitivity, and autoconversion rates, all of which intensify cloud albedo and contribute to a stronger cooling effect over continental regions.

Regions of high anthropogenic activity since the pre—industrial era, such as Europe, Asia, and America, exhibit the strongest (most negative) ERF (Figure 2a), as expected. In Europe, the PPE mean ERF is -3.96 W m$^{-2}$, primarily driven by ACI due to a high anthropogenic sulfur emission, with minor ARI contributions. Notably, the uncertainty in the ERF over Europe is relatively low, at 0.41 W m$^{-2}$ (a 5-95 percentile range of 39%). Asia also shows a substantial ERF, over -4 W m$^{-2}$ on average, largely due to ACI in China, which alone averages over -9 W m$^{-2}$. Furthermore, ARI has a significant contribution to the total ERF over China as well. The ERF uncertainty over Asia has an uncertainty of 110%. Additionally, other regions characterized by high ERF values and significant uncertainty can be found in South America and Africa near industrialized anthropogenic and biomass burning emission sources (Figure 2). Sections 3.1.1 and 3.2 provide a more detailed examination of the factors contributing to these uncertainties.

The ERF over the North Pacific Ocean is the highest of all the oceans at -5.8 W m$^{-2}$, accompanied by a significant uncertainty of 0.58 W m$^{-2}$ (with a 5-95 percentile range of 147%). Structural cloud biases in ECHAM6-HAM cause the strongest



**Figure 1.** Global annual mean probability density functions between 1850 to 2010 of (a) effective radiative forcing (ERF), (b) ERFaci, (c) ERFari from perturbing aerosol parameters. 3 million model emulator-derived variants are used. Our 90% credible range is shown by the black line with the 'X' marker presenting the median. Other estimations shown are using 90% credible ranges, with their marker representing their respective medians (Myhre et al., 2013; Yoshioka et al., 2019; Bellouin et al., 2020; Regayre et al., 2018; Smith et al., 2021; Albright et al., 2021; Forster et al., 2021). The two perturbed parameter ensemble papers, key motivators here, are highlighted in bold. For Shindell et al. (2013b), the 5th to 95th percentile values are calculated as multimodel mean plus and minus standard deviation times 1.645.

.





marine ACI forcing (the North Pacific of around -4 W m$^{-2}$) to differ from Regayre et al. (2018); Smith et al. (2020) and Shindell et al. (2013b) with the strongest forcings over the regions of persistent stratocumulus clouds. High-latitude marine

regions demonstrate considerable uncertainty in the ERF due to aerosol-cloud interactions, while tropical marine regions show comparatively smaller uncertainties (Figure 2d). Southern Asia and the eastern Pacific exhibit high ERF values that are consistent with the projections of CMIP6 models. Smith et al. (2020) suggests those heightened ERF values are primarily driven by negative changes in cloud cover in the respective CMIP6 models. Although the structural cloud bias in ECHAM6-HAM reduces the extent of negative cloud cover change (Neubauer et al., 2019), thereby mitigating the ERF in our model compared

to other PPEs, such as HadGEM (Regayre et al., 2018).

The ERFari in the PPE of Yoshioka et al. (2019) has a 5-95% percentile range of -0.16 to 0.11 W m$^{-2}$, and hence our PPE shows a more negative ERFari with a larger uncertainty (5-95 percentile range between -0.48 and -0.09 W m$^{-2}$ ). However, as noted by Mulcahy et al. (2020), the HadGEM and UKESM1 models have structural issues that may explain the differences from our ERFari estimates. For example, the UKESM1 has an underestimated black carbon mass contributing, which can

partly contribute to these differences (Mulcahy et al., 2020). The PPE ERFari uncertainty range estimate here is within the AR6 range from Forster et al. (2021). The latest IPCC AR6 report indicates a very low likelihood of a positive ERFari. Our uncertainty range for the ERFari aligns closely with IPCC AR6.

### 3.1.1 Parametric Uncertainties in Aerosol ERF

It is important to evaluate the causes of aerosol ERF uncertainty at the global mean and regional scales. Figure 3 shows regional

parameter contributions to the ERF uncertainty(Figure 3a), ACI (Figure 3b), and ARI (Figure 3c). Additionally, Table 2 presents the percentage contribution of each parameter to the global ERF uncertainty depicted in Figure 3a.

Among the top ten parameters contributing to ERF uncertainty, five are related to sulfate and two from biomass burning. The sources of uncertainty in aerosol ERF identified here, surprisingly, due to large differences in model choices, overlap closely with causes found in Yoshioka et al. (2019), Regayre et al. (2018), and Regayre et al. (2023). For example, natural

aerosol emissions contribute significantly to both aerosol ERF and ERFaci uncertainty. The parameters of global importance are mostly consistent with Yoshioka et al. (2019), in that EMI_ANTH_SO$_2$, EMI_DMS, and EMI_CMR_BB are all in the top 5 contributing parameters to ERF uncertainty. We find that ERF uncertainty (Figure 3a) is largely determined by ACI uncertainty (Figure 3b), with a few contributions from ARI (Figure 3c). We also find that there are large regional variations in causes of uncertainty, similar to Regayre et al. (2014), such as EMI_BB, EMI_DMS, and EMI_FF, with these parameters contributing

largely to global uncertainty as in Regayre et al. (2018). While we do not consider perturbation of cloud parameters, much of the HadGEM aerosol contributions to ERF uncertainty are consistent with our study. The considerable overlap in key causes of ERF uncertainty is surprising, as different models have large structural differences that distinctly affect the distribution of clouds and moisture. This presents strong motivation for a future multi-model PPE using similar parameters to better understand the structural uncertainties between more models with distinctly different model structures.

There are differences in the magnitude of ERF uncertainties between ECHAM6-HAM and HadGEM linked to natural aerosol parameters, such as sea salt emissions. These different sensitivities likely arise from the considerable structural differ-





**Figure 2.** Annual emulated mean (a) ERF, (c) ERF ACI, (e) ERF ARI and 2 * standard deviation (2σ) (b) ERF, (d) ERF ACI, (f) ERF ARI. 3 million model variants are used to calculate the mean and standard deviation.

ences between the two models. For instance, HadGEM uses the Gong (2003) parameterization, which lacks a sea temperature dependence. In contrast, ECHAM6-HAM uses the Long et al. (2011) parameterization, with a sea surface temperature dependence from Sofiev et al. (2011). This structural difference can produce very large differences in the spatial distribution of sea

salt emission and therefore aerosol, as shown in Venugopal et al. (2025).





Most often, parameters that cause global mean uncertainty produce some levels of uncertainty across most regions. For example, EMI_CMR_BB produces a global ERF uncertainty of 13% but varies region-to-region from 22% to 4%, which is consistent with Regayre et al. (2018). The largest contribution to the global ERF parametric uncertainty in ECHAM6-HAM comes from the fossil fuel emission parameter (EMI_FF), which accounts for 19% (Table 2 and Figure 3a) of the ERF un-
certainty, and anthropogenic sulfate emissions (15%). Other significant sources of uncertainty to ERF include emissions of carbonaceous materials by biomass burning (EMI_BB; 11%; refer to Table 2) and the size of emitted biomass burning particles (EMI_CMR_BB, 13%). In total, sulfate-related parameters (EMI_FF, $SO_2$Reactions, EMI_ANTH_SO2, EMI_DMS, SO4_COATING, KAPPA_SO4, EMI_CMR_FF) contribute 57% to the aerosol ERF uncertainty (Figure 3a), whereas biomass burning related parameters (EMI_CMR_BB, EMI_BB, BC_RAD_NI) make up 27.5% of the aerosol ERF uncertainty (Fig-
ure 3a). Europe is particularly sensitive to sulfate parameters, which are the primary contributors to ERF uncertainty in this region (73%; Figures 3a). These findings align with the conclusions of Thornhill et al. (2021), which highlight the substantial impact of $SO_2$ emissions on the global ERF. Here, we also highlight the great importance of capturing sulfate parameters, including the sulfur chemistry cycle. Regayre et al. (2015) suggests that due to the relatively short lifespan of aerosols in the troposphere, the influence of aerosol parameters on uncertainties in aerosol radiative forcing is strongly regional.

Biomass burning contributes to a significant proportion of emissions in Africa, South America, and Australia (van Marle et al., 2017; Li et al., 2024) (Figure 3). Over Africa, 50% of the ERF uncertainty comes from biomass burning parameters, mostly from ACI (Figure 3b). For the ARI uncertainty over Africa, 23% is from the imaginary refractive index of black carbon (Figure 3c). Although Tropical and South Atlantic Oceans have no biomass burning sources, they have a substantial contribution of biomass burning to the uncertainty in ERF (35% and 25%, respectively) related to outflow from (primarily)
Africa (Figure 3a and Figure 2a). The regional dependence of ERF on these carbonaceous aerosol parameters is consistent with the findings of Regayre et al. (2018).

Natural aerosol emissions (EMI_DMS (10%), EMI_SS (6%), and EMI_CMR_BB (13%)) are an important source of ERF uncertainty to ECHAM6-HAM. They contribute over 29% to the total aerosol ERF uncertainty and cause even greater uncertainties to the ERFaci (36% Figures 3b). The importance of these emission parameters to the ECHAM6-HAM ERF aligns with
Carslaw et al. (2013), Regayre et al. (2018), and Regayre et al. (2023), though DMS causes more aerosol ERF uncertainty than sea salt in our PPE. The contribution of dust emissions to ERF uncertainty is minimal. This finding aligns with Thornhill et al. (2021), who also concluded that dust has a limited impact on ERF.

### 3.1.2 The role of the minimum CDNC in ECHAM6-HAM

Uncertainties shown in Figures 2c and 2d only include the parametric uncertainties and exclude structural model uncertainties.
Structural limitations, such as missing or oversimplified aerosol processes, can introduce biases and significantly alter model simulations (Regayre et al., 2023). For instance, ECHAM6-HAM uses a minimum Cloud Droplet Number Concentration ($CDNC_{min}= 40$ cm$^{-3}$) in order to be in line with top-of-atmosphere energy considerations (Lohmann and Neubauer, 2018; Neubauer et al., 2019). The fact that such a large value of $CDNC_{min}$ is needed suggests it is compensating for structural error





**Figure 3.** 200,000 model variants and their respective contribution of uncertainties from individual parameters in (a) ERF, (b) ERFaci, (c) ERFari. Only parameters causing at least 5% of the total uncertainty are shown. No hatches represent parameters perturbing aerosol emissions. Diagonal hatching represents microphysics parameters, crossed hatches represent chemistry perturbing parameters, and horizontal hatches show the emitted diameter-sized parameters.





in the model, either in the activation scheme or in terms of missing aerosol species or processes. Also, there may be structural
uncertainties in ice crystal nucleation in cirrus clouds or the stratocumulus cloud cover (Neubauer et al., 2019).

The baseline version of ECHAM6-HAM uses an (unphysical) lower limit of $40 \ cm^{-3}$ for CDNC ($\mathrm{CDNC_{min}}$) to compensate
for undiagnosed structural model inadequacies. If we treat $\mathrm{CDNC_{min}}$ as an uncertain parameter with a range between 1-40
$cm^{-3}$, the mean ERF shifts more negative (strengthens by 29%) and aerosol ERF uncertainty grows to between -2.22 to -
1.08 W $m^{-2}$. The reason for this shift is that all values of $\mathrm{CDNC_{min}}$ are lower than the default value of $40 \ cm^{-3}$. Hence,
the enhanced effects of increased aerosol on CDNC, from anthropogenic emissions since the Pre-Industrial, will be larger in
the present day as the baseline CDNC of $40 \ cm^{-3}$ will be less sensitive to anthropogenic aerosol than when CDNC is below
$40 \ cm^{-3}$ (e.g. Hoose et al., 2009). So, the control ECHAM6-HAM ERF is kept artificially weaker by using the $\mathrm{CDNC_{min}}$ as
a compensating adjustment for structural model inadequacies (Neubauer et al., 2019; Hoose et al., 2009; Zhang et al., 2012;
Lohmann and Neubauer, 2018). The individual parameter contributions to ERF when $\mathrm{CDNC_{min}}$ is included in the variance-
based sensitivity analysis are shown in Figure S3.

For a future PPE, including a perturbation of parameters in the activation scheme, we propose to impose no (or a very small)
limit on CDNC and constrain the PPE with observations of both CDNC and aerosol parameters (including proxy for CCN),
to achieve observationally constrained aerosol ERF that may overcome the need to adjust for structural model features. We
expect this future work will constrain natural emission parameters towards higher values, leading in-turn to a lower CDNC and
weaker aerosol ERF, in line with Regayre et al. (2023).

## 3.2 Comparison to observations and causes of present-day uncertainty and bias

When directly comparing our PPE with the RemoTAP aerosol retrievals from the PARASOL satellite (Hasekamp et al., 2024),
ECHAM6-HAM generally underestimates AOD (Figure 4a), overestimates AE (Figure 4b), and only slightly underestimates
SSA (Figure 4c). This suggests that the aerosols produced in ECHAM6-HAM are too few, too small, and slightly too scattering.
Although the bias in scattering and absorption of aerosols varies significantly over different regions (Figure 4c). Quantifying
parametric uncertainties from the PPE may help identify parameters to mitigate this aerosol model bias. For example, Figure 5
shows the high contribution of natural aerosol emissions to global uncertainties of AOD, AE, and SSA. The PPE means, bias,
and uncertainties for AOD, AE, and SSA are shown in Figures 6, 7, 8. If a region of the model is biased but still within the
uncertainty range, constraining parameters could be used to reduce the bias, as in (Regayre et al., 2023).
It is important to note that the POLDER data from the Arctic and Antarctic regions are biased and excluded from this
comparison (Hasekamp et al., 2024). The uncertainties from POLDER are presented in the relevant sections when compared
to the PPE, from Hasekamp et al. (2024). For this comparison, the model PPE has been co-located in space and time with the
POLDER satellite observations. For more detail, Figures S4, S5, and S6 in the Supplementary Information show global maps
with the contribution of each parameter to the model uncertainty in AOD, AE, and SSA, respectively.







**Figure 4.** Regional direct comparison between the mean (a) Aerosol Optical Depth, (b) Ångström, and (c) Single Scattering Albedo observation from POLDER (red dot) with the box and whisker distribution from the PPE. The box represents the interquartile range (IQR; Q1 25th to 75th percentile of the PPE distribution), with the red vertical line representing the median of the PPE. The whiskers are defined by Q1 − $1.5 \times$ IQR and Q3 + $1.5 \times$ IQR. Uncertainties from POLDER over land for (a) AOD is $\pm 0.04$, (b) AE is $\pm 0.4$, and (c) SSA is $\pm 0.03$. The uncertainties from POLDER over the ocean for (a) AOD is $\pm 0.03$, (b) AE is $\pm 0.25$, and (c) SSA is $\pm 0.03$.




**Figure 5.** 200,000 model variants and their respective contribution of uncertainties from individual parameters in present-day (a) AOD, (b) AE, (c) SSA. Only parameters above 5% are shown. No hatches represent parameters perturbing aerosol emissions. Diagonal hatching represents microphysics parameters, crossed hatches represent chemistry perturbing parameters, and horizontal hatches show the emitted diameter-sized parameters.




**Table 2.** Percentage contribution to global annual mean ERF uncertainty by individual parameters over 200,000 model variants.

| Parameter | Uncertainty Contribution (%) |
|---|---|
| EMI_FF | 19.25 |
| EMI_ANTH_SO2 | 14.75 |
| EMI_CMR_BB | 12.66 |
| EMI_BB | 11.33 |
| EMI_DMS | 9.53 |
| SO2_REACTIONS | 8.13 |
| EMI_SS | 5.98 |
| NUC_FT | 4.16 |
| EMI_CMR_FF | 3.70 |
| BC_RAD_NI | 3.55 |
| DU_RAD_NI | 2.45 |
| DRYDEP_ACC | 1.26 |
| SO4_COATING | 1.05 |
| EMI_DUST | 0.97 |
| KAPPA_SO4 | 0.41 |
| DRYDEP_COA | 0.25 |
| WETDEP_IC | 0.21 |
| EMI_BF | 0.12 |
| KAPPA_SS | 0.09 |
| EMI_CMR_BF | 0.07 |
| DRYDEP_AIT | 0.04 |
| WETDEP_BC | 0.02 |
| PH_PERT | 0.02 |

### 3.2.1 Present-day AOD

The PPE mean co-located global average AOD is 0.12 (Figure 6b and Figure 4a). The PPE has a 5-95% percentile range between 0.09 and 0.16, whereas the POLDER mean of 0.15 ($\pm$ 0.04 over land; $\pm$ 0.03 over ocean; Hasekamp et al. (2024)) is close to the high end of the PPE range (see Figure 6a and Figure 4a). Global averages of AOD in the PPE are underestimated compared to observations (Figure 6d and Figure 4a). The largest statistical differences between the model and observations are primarily found over land, where the model tends to underestimate AOD (Figure 6b). The few regions of positive bias over land are found over China and Northern Africa. Over the ocean, there are more regions with positive AOD bias (Figure 6b), but at higher latitude oceans, there is a significant negative bias (Figure 6b). Regions with high absolute AOD uncertainty correspond to areas with significant differences between the model and observational data, as shown in Figure 6d. This observation is consistent with comparisons to MODIS AOD (Figure S7).





Differences between the model and observations in Figure 6b are most negative over Africa, India, and North America, which are major dust sources. Additionally, AOD over China/East Asia is positively biased in ECHAM6-HAM (Figure 6b). Generally, marine regions have larger AOD values in the PPE than in observations, except the Southern Ocean, which is negatively biased.

        Dust is a major source of uncertainty in our PPE, as shown in Figure 5a. Dust is a known bias for larger aerosol particles
in the ECHAM6-HAM model (Tegen et al., 2019). The uncertainty in Africa primarily arises from dust and biomass burning emissions (EMI_DUST and EMI_BB), as illustrated in Figure 5a. Notably, the bias observed over Africa (Figure 6b) is greater than the uncertainty indicated by our PPE (Figure 6d). To address this bias, we would need to consider additional parameters, such as dust size, which are not included in this analysis.

        Over East Asia, there is a positive bias in AOD (Figure 6b), consistent with findings by Salzmann et al. (2022). This bias
is attributed to a diverse range of aerosol species from anthropogenic sources, with sulfate parameters being the primary contributors (Yang et al., 2024). The substantial uncertainty in our PPE over China (Figure 6d) may offset this bias, as the AOD uncertainty is comparable in magnitude to the bias. This uncertainty is mainly due to sulfate-related parameters, including EMI_ANTH_SO$_2$, Kappa_SO$_4$, SO$_2$Reactions, EMI_FF, and DRYDEP_ACC (see Figure 4 and Figure 5a). The bias in this region has been previously identified by Tsikerdekis et al. (2023). Over this region, increasing the scale factors for SO$_2$Reactions
and KAPPA_SO$_4$ enhances AOD uncertainty (Figure 5a and Figure S4) through increasing sulfate hygroscopicity and lifetime, exacerbating existing model overestimates in hydrophilic aerosol growth (Tsikerdekis et al., 2023). These parameters probably can compensate to some extent for the relative humidity bias found by Tsikerdekis et al. (2023) in structural uncertainty.

        Oceanic regions that have high AOD uncertainty coincide with regions of positive AOD bias are shown in Figure 6c. Salzmann et al. (2022) attributes this bias to the lack of shallow convective clouds precipitating in ECHAM6-HAM, therefore
resulting in a lack of wet removal of hydrophilic aerosol by these clouds, which are frequent in these regions. Our PPE also identifies this bias through parametric uncertainties (Figure 6d), presenting a possibility which may compensate for this structural inadequacy related to the convective scheme (Salzmann et al., 2022). Additionally, Tsikerdekis et al. (2023) identified that the regions of heightened relative humidity bias coincide with the oceanic regions in our PPE that are both positively biased (Figure 6c) and have notably large AOD uncertainty (Figure 6e). As a result, excess modeled relative humidity enhances hy-
groscopic aerosol growth, increasing AOD without the important convective removal mechanism. Our PPE suggests that this growth primarily comes from sea salt aerosols, as shown in Figure S4. Interestingly, these regions of high absolute uncertainty are not as prominent in the relative uncertainty field (Figure 6e). This suggests that absolute uncertainty is more sensitive to the magnitude of AOD itself, whereas relative uncertainty highlights regions where uncertainty is large relative to AOD. This distinction indicates that different processes may control absolute and relative uncertainty, particularly over low AOD regions
such as the Southern Ocean, emphasizing the need to assess both metrics when evaluating model uncertainty and observational constraints.

        Figure 5a shows that sea salt and DMS emissions cause the highest uncertainties in AOD (37%). This is partly because sea salt and DMS emissions have much wider areas to emit and thus contribute to AOD uncertainty over most regions (Figure S4). The parameters that control much of the AOD uncertainty over land (Asia and Europe) mostly are from anthropogenic activities





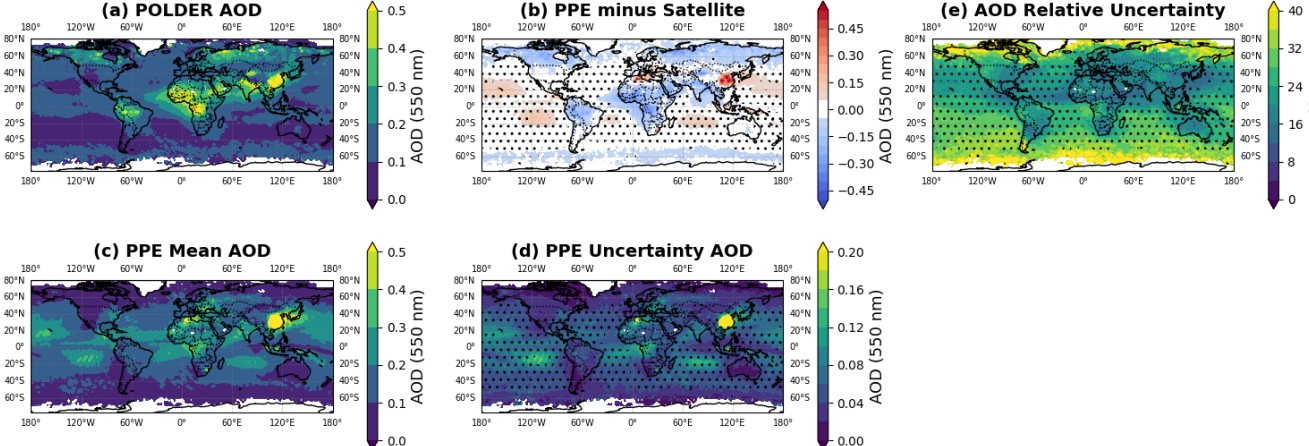

**Figure 6.** Aerosol Optical Depth (AOD 550 nm) global (a) POLDER instrument measurement, (b) difference between the PPE and POLDER, (c) PPE mean, (d) PPE absolute uncertainty (2$\sigma$), and (e) PPE relative uncertainty during 2010. PPE simulations are co-located to the same coordinate and time as POLDER. Stippling indicates where the difference is not statistically significant at the (95% level of confidence, Student's t-test).

(Figure 3a). Depositional parameters, such as WETDEP_IC and DRYDEP_ACC, have a globally uniform contribution to the uncertainty (of around 11% and 8%, respectively) across all regions.

Figures 6c and 4 show a slightly lower emulated AOD than that observed over the Southern Ocean. The relative AOD uncertainty over the Southern Ocean is higher than any other region (Figure 6d). The high relative importance of sea spray and DMS emissions aligns with Yoshioka et al. (2019) and Carslaw et al. (2013), which highlight the substantial uncertainty
associated with natural aerosol emissions, also present spatially in the ECHAM6-HAM model. Excluding Antarctica, the relative uncertainty in AOD over the Southern Ocean is 33%, higher than the global average of 23%. Sea spray and DMS emissions cause around 45% of the Southern Ocean AOD uncertainty.

Overall, our results reveal that present-day AOD (Figure 5a) and ERF (Figure 3a) have three common sources of uncertainty in the top seven uncertainties (EMI_DMS, EMI_SS, SO$_2$Reactions). This is consistent with the work of Regayre et al. (2018),
which suggests that although both AOD and aerosol forcing have some influence from common aerosol properties, their main sources of variability differ due to distinct processes, as found in Lee et al. (2016). For example, sea salt emissions cause significant perturbation to AOD uncertainty, whereas sea salt emissions only perturb ERF uncertainty by 6%. Conversely, biomass burning-related parameters cause substantial perturbation to the ERF uncertainty (Figure 3), but not so for AOD uncertainty (Figure 5). Thus, observational constraints to match present-day AOD will not guarantee constraints to all parameters relevant
to ERFaci, even though they may still provide a useful constraint on ERFari (Watson-Parris et al., 2020).





### 3.2.2 Present-day Ångström Exponent

The global Ångström Exponent (AE) from the ECHAM6-HAM PPE exhibits an overestimation relative to PARASOL observations, particularly over oceanic regions and high latitudes (Figures 7b and 4b). This suggests that particle sizes in the ECHAM6-HAM model are too small relative to observations. The model particularly underestimates AE over dust emissions
in Northern Africa and over parts of East China, although it should be noted that POLDER AE has a positive bias over Dust regions (Hasekamp et al., 2024). Strong positive AE bias is found over the Southern Ocean, the Americas, and large parts of Asia and Australia.

Uncertainty attribution in Figure 5b reveals that key parameters controlling AE uncertainty globally are mostly EMI_DMS, EMI_SS, SO$_2$Reactions, and NUC_FT. Some parameters, such as SO$_2$Reactions, NUC_FT and EMI_BF contribute almost
uniformly to AE uncertainty in all regions. These drivers contrast with AOD uncertainty (Figure 5a), which is primarily linked to perturbations in aerosol mass emissions, highlighting the distinct sensitivity of AE to processes affecting particle size and mixing state.

Regions with the largest AE biases, such as high latitudes, coincide with high parametric AE uncertainty (Figure 7d). Over the regions of AE uncertainty larger than 0.1 (Figure 7c), SO$_2$Reactions, SO4_Coating, EMI_DMS, and EMI_SS collectively
account for almost 50% of the uncertainty (Figure 5b). The dominance of sulfate-related parameters underscores the critical role of aerosol aging and growth related processes in modulating size distributions. Over land, SO4_Coating, EMI_DUST, and SO$_2$Reactions drive uncertainty, whereas oceanic regions are influenced most by sea salt emissions (EMI_SS) and DMS oxidation pathways (SO$_2$Reactions and EMI_DMS) (Figure 5b; Figure S5).

Particle aging (SO4_Coating) has a particularly large regional impact on AE uncertainty (Figure 5b) as sulfate coatings
modify particle hygroscopic growth. Sulfate coatings (SO4_Coating) enhance particle hygroscopicity, accelerating water uptake and growth of fine-mode aerosols, which systematically reduces AE in regions like the Southern Ocean, and thus increases its parametric uncertainty (Figure 5b). The overlap of SO$_2$Reactions in both AOD and AE uncertainties (Figure 5) highlights the important role of sulfate in both mass and size regulation. SO$_2$Reactions and EMI_SS influence the mixing state and hygroscopic growth of aerosol processes, which are critical for determining the aerosol optical properties. Tegen et al. (2019)
and Salzmann et al. (2022) emphasize that the underrepresentation of coarse-mode particles, especially sea salt, can lead to substantial overestimations of AE, which is present in the comparison with the PARASOL AE (Figure 7b).

The interplay between model biases (Figure 7b) and parametric uncertainty (Figure 7d) is amplified in the Southern Ocean, due to the lack of coarse model sea salt particles, which are strongly present over the Southern Ocean (Venugopal et al., 2025). A pronounced positive AE bias (Figure 7b) aligns with an underestimation of coarse-mode sea salt aerosols. This corresponds
to a lower AOD (Figure 6c), suggesting that a reduction in aerosol mass may influence AE and AOD biases. Given the region's high proportion of sulfate and sea salt, uncertainties in EMI_SS, EMI_DMS, and SO$_2$Reactions (Figure 5b) likely propagate through both size distribution and mass, amplifying uncertainty (Figure 6c). This is consistent with global model challenges in representing marine aerosol emissions and aging (Tegen et al., 2019). Reducing these uncertainties will require improved constraints on sea salt flux parameterizations and DMS-to-sulfate conversion pathways.



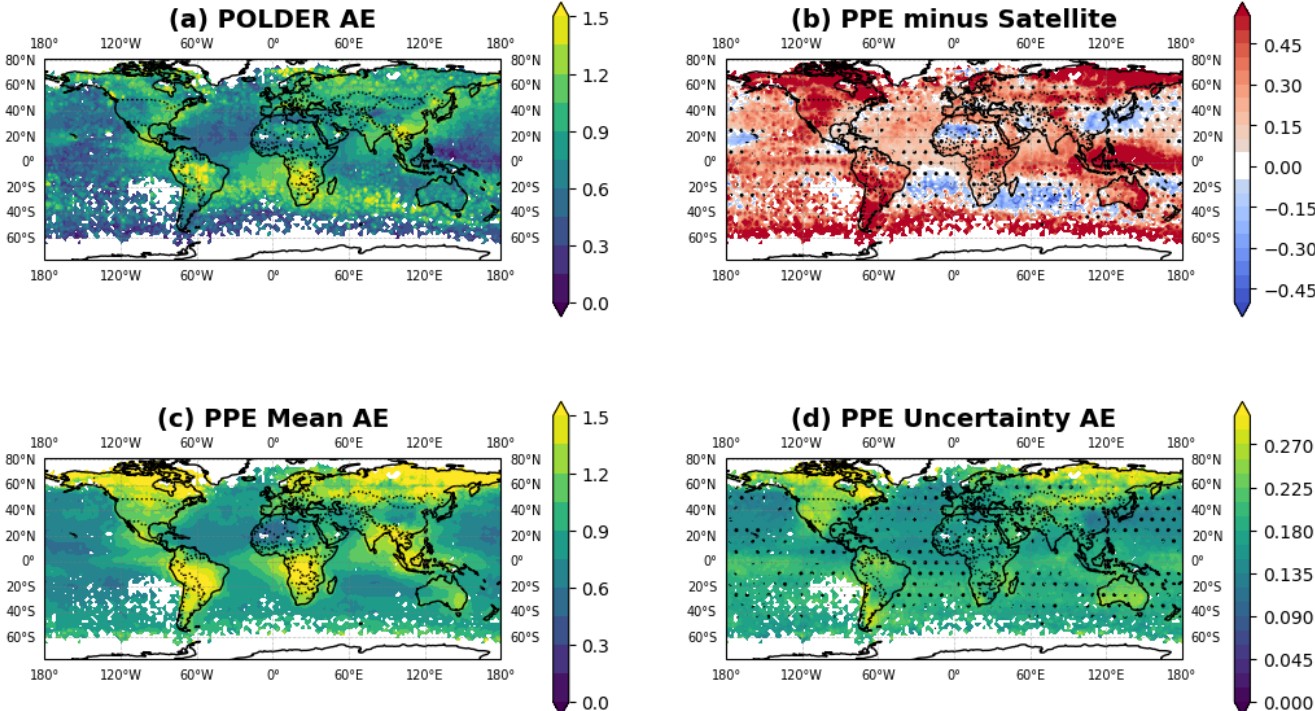

**Figure 7.** As in Figure 6, but showing Ångström Exponent.

In contrast, North Africa exhibits a negative AE bias (Figures 7c and 4b), suggesting the model overestimates dust aerosol size. This conflicts with prior ECHAM6-HAM validations (Tegen et al., 2019), and might be related to a positive AE bias in PARASOL (Hasekamp et al., 2024). Our PPE mean AE (0.67) here aligns with earlier studies, yet the region shows large uncertainty tied to EMI_DUST (Figure 5b and Figure S5). The similarity in magnitude between the AE bias and the uncertainty associated with dust emissions suggests that perturbations in EMI_DUST could account for the observed bias.

ECHAM6-HAM exhibits a strong positive AE bias over the Intertropical Convergence Zone (ITCZ) and the South Pacific Convergence Zone (SPCZ; Figure 7c). This aligns with previously identified precipitation biases linked to cloud cover and ice water path uncertainties (Neubauer et al., 2019; Stevens et al., 2013), as structural uncertainties in cloud parameterization and parametric drivers like WETDEP_IC amplify biases by reducing coarse-mode sea salt via excessive wet deposition. Furthermore, EMI_SS and EMI_DMS uncertainties disrupt sulfate-driven aerosol growth (Figure S5; Croft et al., 2009). The model's underestimation of sea salt particle size, dominant in these regions, exacerbates the AE overestimation (Figure 7c), as DMS-derived sulfate condenses onto fewer coarse particles (Salzmann et al., 2022).





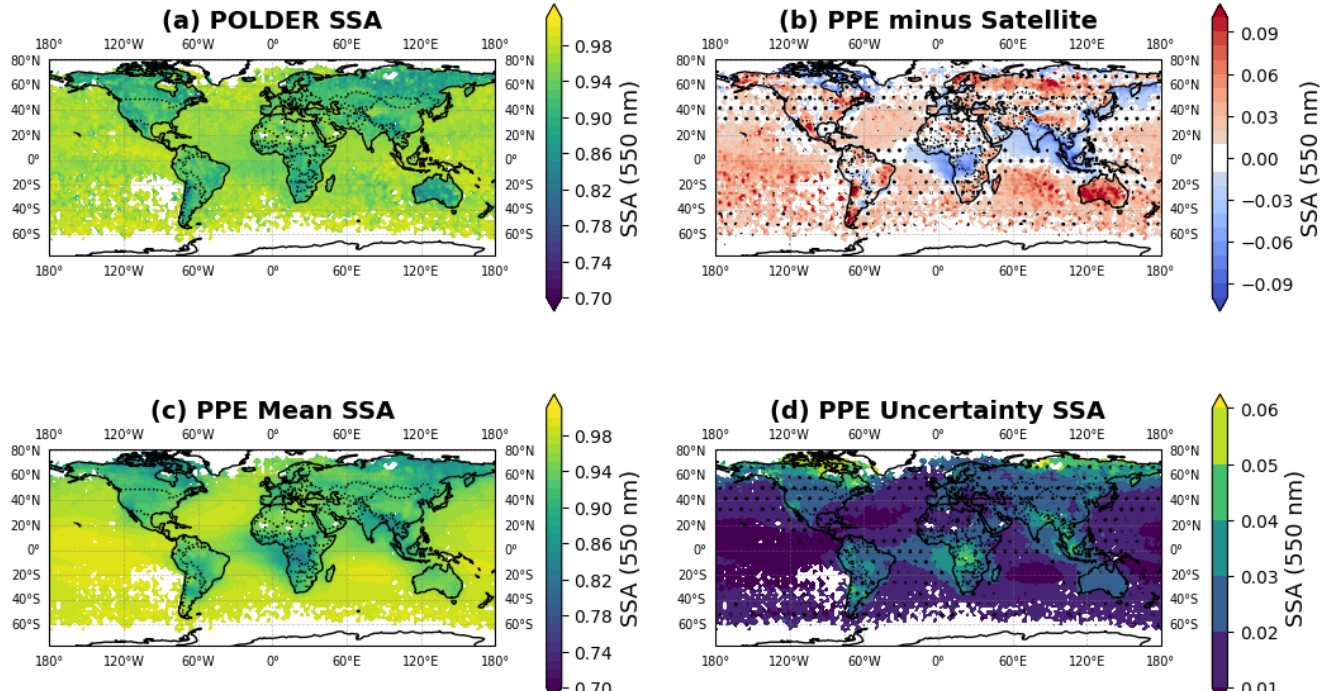

**Figure 8.** As in Figure 7 but showing Single Scattering Albedo.

### 3.2.3 Present-day SSA

The global PPE reveals a systematic overestimation of single-scattering albedo (SSA) over oceans relative to PARASOL observations (Figures 8b and 4c), indicating that modeled marine aerosols are less absorptive than from PARASOL retrievals. The model presents a negative SSA bias around central Africa and parts of South East Asia and India, whereas a strong positive bias is present over Australia and parts of the Arctic region.

SSA uncertainty from the PPE (Figure 8d) is insufficient to account for the strong positive bias over the oceans and Australia. However, in regions with negative SSA bias (such as central Africa, South East Asia, and India, Figures 8b), the model uncertainty is larger than the bias itself; therefore, the observed underestimation falls within the ensemble's range of variability.

Uncertainty attribution identifies BC_RAD_NI as the largest global contributor to SSA uncertainty (Figure 5c), reflecting its direct control over BC absorption efficiency. Secondary drivers include EMI_DMS, EMI_SS, and $SO_2$Reactions, particularly over the Southern Ocean, where marine aerosol processes outweigh BC's influence (Figure 5c, Figure S6)). Across most land regions, BC_RAD_NI and DU_RAD_NI account for approximately 20% and 15% of SSA uncertainty, respectively (Figure 5c). These parameters govern the imaginary refractive indices of black carbon (BC) and dust, directly influencing their absorptive properties.



SSA uncertainty over the ocean stems from marine emissions (e.g., EMI_SS, EMI_DMS), whereas terrestrial uncertainty is dominated by biomass burning (EMI_BB and BC_RAD_NI), dust (EMI_DUST and DU_RAD_NI), and sulfate chemistry (SO$_2$Reactions) (Figure 5c). The parametric uncertainty in SSA over land is roughly twice as large as over oceans (Figure 8d and Figure 4b). Much of the BC_RAD_NI contributions over ocean are related to outflow of existing aerosol emissions from the land (Figures 5c, Figures S6). Over regions like South America and Africa, where biomass burning is a dominant aerosol source, a negative SSA bias (Figure 8c) may suggest modeled BC is absorbing too strongly relative to observations (Salzmann et al., 2022), or has a bias in the ratio between BC and OC (Organic Carbon). These regions also show elevated uncertainty tied to BC_RAD_NI, EMI_BB, SO$_2$Reactions, and DU_RAD_NI (Figure 5c, Figure S6), underscoring the need for better constraints on BC and dust optical properties and combustion emission fluxes.

### 3.2.4 Model development priorities related to Present-Day aerosol biases

Sulfate-related parameters cause 40% of AOD and SSA uncertainty, and 44% for AE uncertainty. For the most part, this is from the shared influence of SO$_2$Reactions and EMI_DMS across AOD, AE, and SSA, highlighting sulfate's key role in present-day aerosol uncertainty. Regional patterns also highlight how key sources of AOD uncertainty cascade to uncertainties in AE and SSA. Over the Southern Ocean, for example, the elevated AOD relative uncertainty stems from sea salt emissions (EMI_SS) and DMS-derived sulfate production (EMI_DMS), which have smaller size distributions than observations (Figure 7b), amplifying AE biases and uncertainty (Figure 7b, d). Here, marine emissions (EMI_SS, EMI_DMS) drive both SSA and AE uncertainties, while their underrepresentation also reduces AOD (Figure 6c). As a result, we suggest improvements in the treatment of sulfate and marine aerosol emissions (e.g., EMI_DMS, EMI_SS) with a focus on particle size distributions to reduce simultaneous AOD, and AE biases. Additionally, future work with subdivide sulfate-related parameters further to better capture and constrain sulfates global uncertainty sources.

Parametric uncertainties may be able to compensate for some structural uncertainties in present-day aerosol. For example, relative humidity causes problems to aerosol growth in ECHAM6-HAM, relative to observations (Tsikerdekis et al., 2023). Additionally, the model has an AOD bias over marine regions from structural problems related to the convective scheme (Salzmann et al., 2022). Through perturbing sulfate and sea salt parameters, aerosols may grow over China and marine regions enough to compensate for the AOD bias. Additionally, the negatively biased modeled SSA over central Africa overlaps with the PPE uncertainty from BC_RAD_NI, enabling the possibility of mitigating SSA bias through aerosol refractive index parameters. Structural uncertainties in cloud parameterization may be compensated by depositional parametric drivers, like WETDEP_IC, which affect large aerosol particles. Parameters related to sulfate, BC_RAD_NI, and wet deposition may be important for future tuning, as they could partially offset known structural deficiencies related to humidity and cloud interactions.

AE uncertainties from the PPE have very little overlap with the modeled bias relative to POLDER. Some parameters, such as NUC_BL and EMI_BF have global contributions to AE uncertainty. Many regions with the largest AE biases, such as high latitudes, coincide with high parametric AE uncertainty, but the extent is insufficient to account for the AE bias. The cause of this mismatch is likely from structural uncertainty stemming from issues (a lack of) in the larger aerosol size distributions, as shown by Tegen et al. (2019). Perturbing aerosol mass may enhance AE and AOD uncertainties to overlap biases. Therefore,





future PPEs will extend this PPE design to include parameters influencing larger aerosol size modes, possibly through direct perturbation of aerosol mass or size parameters.

## 4    Discussion and Conclusion

Using a perturbed parameter ensemble (PPE) with the ECHAM6-HAM model, we have quantified aerosol-related parametric uncertainties in aerosol effective radiative forcing (ERF) and present-day aerosol properties (AOD, AE, SSA). Our PPE ex-
plores 23 parameters that influence aerosol emissions, removal processes, chemistry, and microphysics, building further on the approach of Yoshioka et al. (2019) developed for the HadGEM model. We also include newly identified parameters informed by recent research and model-specific biases, such as sulfate chemistry. This PPE framework enables us to assess uncertainties at both global and regional scales. Ultimately, our goal is to use PPEs to identify and constrain key ERF causes and uncertainties and to build confidence in climate projections by clarifying the role of parameter uncertainty and their potential to identify
and/or reduce structural model uncertainties. When comparing a PPE across multiple models, structural inadequacies may be overcome by re-tuning model parameters. Despite large substantial structural differences between two models (ECHAM6-HAM and HadGEM), the magnitude and dominant parametric sources of aerosol ERF uncertainty overlap in agreement across the ensembles.

Aerosol-related parametric uncertainties within our PPE lead to an aerosol ERF with a 5-95 percentile uncertainty range
between -1.59 W m$^{-2}$ to -0.89 W m$^{-2}$. Sulfate-related parameters (sulfate chemistry, Anthropogenic SO$_2$ emissions, DMS emissions) cause 57% of global ERF uncertainty, and biomass burning parameters (EMI_CMR_BB, BC_RAD_NI, EMI_BB) 27.5%. While sulfate dominates aerosol forcing in many regions, biomass burning plays a significant role in regions with high biomass burning activity, like Africa. The smaller particle size of black carbon determined from the AE bias, combined with biases in its refractive index (black carbon imaginary refractive index), contributes to an uncertainty in absorption properties
and radiative effects.

Global AOD uncertainties are primarily caused by natural aerosol emissions (DMS and sea salt) (33%) and depositional processes (22%). Sulfate-related parameters contribute over 40% of the uncertainty in present-day AOD, AE, and SSA, underscoring their central role in shaping both aerosol climatology and ERF. Regions with high AOD uncertainty in ECHAM6-HAM often coincide with areas where model biases relative to observations are largest. Many of these regions have a positive AE
bias, suggesting an underrepresentation of coarse-mode particles, as also identified by Tegen et al. (2019). ECHAM6-HAM under-represents coarse-mode aerosols over much of the ocean, particularly the Southern Ocean. Biomass-burning aerosols in ECHAM6-HAM appear smaller and more absorbing than observed by PARASOL, leading to a positive SSA bias, particularly over Africa and South America, although it should be noted that PARASOL has considerable SSA observational uncertainty.

The main causes of uncertainty in ERF do not show full overlap with those in present-day AOD, AE, and SSA. On the
other hand, the different causes of uncertainties between AOD, AE, and SSA in different regions suggest that spatially resolved measurements of these three aerosol properties together can help in constraining parameters that dominate ERF uncertainty, depending on the accuracy of the observations. However, some important causes of ERF uncertainty (most notably the emitted



particle sizes of BB and FF) do not show similar causes of uncertainties in AOD, AE, or SSA. Adding observations of CDNC in addition to aerosol properties is expected to help constrain those parameters, which strongly impact ACI (McCoy et al., 2020).

Future work will aim to constrain the ECHAM6-HAM ERF based on the recent satellite retrievals from the PACE (Werdell et al., 2019; Hasekamp et al., 2019a; Fu et al., 2025) and EarthCARE (Illingworth et al., 2015) missions launched in 2024, which provide the relevant aerosol and cloud observations with unprecedented accuracy.

Many of the model biases that appear when comparing with observations are caused by a combination of parametric uncertainties and structural model uncertainties. The latter may be partially mitigated through parameter tuning; however, this

compensation may mask rather than resolve model deficiencies. To deepen our understanding of the different structural model uncertainties, a multi-model PPE is needed. The tight clustering of ERF uncertainty sources across models suggests that efforts to constrain ERF may benefit from cross-model coordination, especially around sulfate chemistry, DMS emissions, and biomass burning representation. By applying similar perturbations across multiple models (e.g., ECHAM6-HAM, HadGEM, ICON-HAM, EC-Earth, and NorESM), it would be possible to isolate uncertainties stemming from missing processes and

those arising from poorly constrained parameter values. A multi-model PPE is important for CMIP7 to address the model structural uncertainties' contribution to ERF uncertainties and provide a more robust calculation for ERF. Furthermore, this would provide actionable information for improving structural representations of key aerosol processes, leading to improved model skill at simulating climate change.

## 5 Data Availability

All model simulation data are archived and are available by contacting the corresponding author. Parameter values are found in Zenodo, and annual mean PPE diagnostics are available in https://doi.org/10.5281/zenodo.15640132 (Bhatti et al., 2025). POLDER observations can be found in https://public.spider.surfsara.nl/project/spexone/others/PARASOL/DATA/POLDER_ 1.0x1.0_basedon0.1_NPge2/2010/. Additionally, the POLDER measurements combined with AOD, AE, and SSA co-located PPE data can be obtained in Zenodo https://doi.org/10.5281/zenodo.15640132. For higher frequency PPE data (monthly, daily,

3-hourly) or additional PPE diagnostics, contact the corresponding author. The ESEm code used to emulate the simulations is in https://doi.org/10.5281/zenodo.5466563 (Watson-Parris et al., 2021).

*Author contributions.* Author contributions. YAB implemented the model setup, performed model simulations, and wrote the manuscript, with assistance from all co-authors. DWP, LAR, HJ, DN, NS, and OPH assisted in technical model setup, methods, and provided expertise in analysis. All co-authors provided expertise in parameter ranges and helped with the model setup.

*Competing interests.* One of the authors is a member of the editorial board of Atmospheric Chemistry and Physics.



*Acknowledgements.* This research was supported by the European Union, Horizon Europe project CleanCloud (GA 101137639). The simulations and data analysis used the Dutch national e-infrastructure with the support of the SURF Cooperative using grant no. EINF-11775/L1.



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
