# Peer review of "Uncertainty in aerosol effective radiative forcing from anthropogenic and natural aerosol parameters in ECHAM6.3-HAM2.3"

_EGUsphere, 2025_

## Referee Comment (RC1)

Notes on Bhatti et al

Uncertainty in aerosol effective radiative forcing from
anthropogenic and natural aerosol parameters in
ECHAM6.3-HAM2.3

This study provides an examination of Effective Radiative Forcing (ERF) uncertainty in the ECHAM6.3-HAM2.3 (hereafter called EC-H) model using a PPE paradigm to examine its sensitivity to a number of uncertain, but important, parameters that govern processes that influence aerosol-cloud, and aerosol-radiation interactions, and aerosol emissions. A Gaussian process emulator is used to expand the number of members of the PPE in order to more accurately evaluate the parameter values entering into the comparison. The results are compared to POLDER-3 data (AOD, SSA and AE). A summary is provided of the important (and less important) parameters driving uncertainty in this model, and those results are compared to similar studies from other models.

I am a climate modeler with a pretty broad range of experience in model development and use, but I do not have as much expertise as some authors of this study in single, specific areas, so my comments are going to be quite general, rather than focusing on specific items in the study. The study has been done carefully and thoroughly, and I have no specific requests for corrections in methodology, or disagreements with conclusions. But I do think revision would be useful to deliver messages more crisply, cleanly, and completely to make it easier for readers to extract information and put their results into context more quickly. I also think the discussion of structural uncertainty and how it relates to model uncertainty is somewhat superficial and it could be improved. I had a few specific questions about results in figures that should be addressed. I am recommending that it be accepted pending what I think are minor revisions, and offer some comments and suggestions below.

Specific comments:

1. Lines 1-68: A first reading of the abstract and section 1 still left me wondering what the precise target was for the study. Could you state the goals a little more crisply? On line 6 you say your goal is to "address a gap" (my condensation of your words). If I read your sentence carefully, I think you are saying that the gap is in quantifying the sources of uncertainty. So is the goal to identify some of the processes that contribute most to uncertainty in EC-H? And then compare those results to other models? You also eventually connect some of your results on parametric uncertainty with known structural deficiencies in the EC-H model. Maybe you could say something like these sentences explicitly, and provide hints on how (or whether) this information can be used to reduce uncertainty? I also disagree mildly with your characterization of what

structural uncertainty is, and how it contributes to model uncertainty (more below). I found the discussion of Regayre et al (2023) to have helped me understand what can and cannot be gleaned about structural and parametric uncertainty from PPE studies. Perhaps a little more of that kind of discussion could be inserted here also, since many authors are shared between that and this study.

2. Lines 10-11: I had a somewhat different interpretation of your results than expressed in this sentence. I think your sentence could be read to indicate that the leading causes of ERF uncertainty are associated with parameter uncertainties. I am not sure this is the case because some of your results indicate that structural deficiencies play a very large role in ERF uncertainty also, and you have by no means investigated many aspects of structural uncertainty. I personally believe structural uncertainty plays a very large role. I do think it is true that of the parametric uncertainties that you examined there were many common features with other PPE studies. Could you make your arguments more compelling or revise the sentence to agree more with my statement?

3. Lines 15-16: states "PPEs can reduce some structural model biases through parameter adjustments, but others persist." Maybe this topic deserves more discussion. Such adjustments might lead to a positive outcome, but isn't this fixing a problem for the wrong reason with the wrong solution? Presumably one doesn't like to correct a problem by adding an arbitrary correction of a (possibly) satisfactory parameterization to produce a reasonable result with a model containing a significant structural deficiency. It is unsatisfactory if the goal is to improve understanding and representation of the underlying physics, and it is at best an expedient kludge to avoid some other problem.

4. Lines 43-45: Identifying structural uncertainty with deficiencies in "coding" feels misleading. It could be that a process treatment represents one process very accurately, but entirely ignores another. Is this a coding problem? It is an "understanding problem" reflecting a lack of understanding or some other choice by the team responsible for treating those processes. Can you replace "coded" with "represented", "treated", "formulated", etc?

5. Lines 54-55: The phrase "before observational constraints" doesn't deliver a clear message. Maybe "alongside observational uncertainties that constrain parameter choices", or some other phrasing.

6. Line 65: It seems to me that the paper has two main goals:
   a. Characterize the uncertainties for the EC-H model
   b. Identify some of the shared parametric uncertainties with other models and PPE studies.

   It might be worth adding a sentence or two here making these points before discussing what is going to be shown in various sections.

7. Lines 103-106: Are you partitioning the ARI and ACI contributions to ERF using clear-sky and all sky fluxes, or some more sophisticated method (e.g., Ghan (2013), or an APRP method)? Please specify.

8. Line 179: perhaps insert "study" after the word "each" and change "sample" to "samples"?

9. Line 189: typo "perturbationsc"

10. Line 203-205.  These lines make explicit that the study is not addressing structural uncertainty. I believe that previous sections of the text were vague on this point, and I also think that the later discussion of the role of the CDNC minimum do deliver messages that are relevant to structural uncertainty.

11. Lines 195-206: I am quite interested in this discussion of common features, conclusions, and differences between the EC-H simulations and analysis done here and studies with other models.  I think the authors did a nice job of summarizing some of those differences, and I hope that there is more discussion like this in the summary sections of the study.

12. Lines 216-220.  Are you saying that the biases in marine stratocumulus in ECHAM6 are structural, unlike those in HADGEM, UKESM, and the models analyzed by Shindell?  I am unclear about the messages in these lines.

13. Lines 220-223. Do you think that the smaller uncertainties you identified in the marine tropics are just due to the crudity of the convective parameterizations and their ACI treatments in all global models, or do you think that those representations are actually robust and we really understand and are representing those processes accurately?

14. Line 233-305: I like the discussion in sections 3.1.1 and 3.1.2 a lot. Would it make sense to change the title of section 3.1.2 to mention "CDNC minimum as an example of Structural Uncertainty" rather than to list the section with the specific topic of minimum CDNC? That way you have a section dealing with parametric uncertainty and another dealing with structural uncertainty (with a big focus on CDNC lower bounds).  As you point out in the first sentence or two of 3.1.2 there are other possible contributions. The lower bound on CDNC just provides a convenient and very easy example to demonstrate the existence of structural uncertainty. It might be useful to list some of the possible explanations for the need for such a limiter (CDNC also depends on estimates of subgridscale vertical velocity and internal circulations within the clouds driving aerosol activation, and unresolved or under-resolved exchanges between cloud free and cloudy air masses at cloud top. Other candidates that occur to me that are driven by radiative cooling, evaporation and drop sedimention. These are processes that occur to me, but you may have a better list).

15. Lines 301-305: I don't really understand the philosophy behind the proposed PPE study described on the lines of this paragraph. Are you just proposing an alternate kludge (to

the CDNC lower bound), to avoid dealing with lack of understanding of the important processes? I could imagine employing a Machine Learning or AI method to connect CDNC, aerosol properties, and meteorology/cloud properties, but those words and methodologies are not mentioned on these lines, and it is hard for me to envision how one would use a PPE to improve structural deficiencies, as hinted at on these lines. Can you be more explicit about what you are proposing, or drop the paragraph?

16. Lines 306-320: I am assuming the black circles on each line (box/whisker plot) of figure 4 represent the value for a quantity and region for a particular member of the PPE. If I am correct, could you please state that in the caption? And if that supposition is correct, why do all the ensemble member values for AOD, AE and SSA cluster at the extreme ranges of the statistics, rather than frequently falling within the 25 and 75 percentile ranges? Since the circles never seem to fall within the interquartile range it suggests my stated supposition is incorrect, or there is a problem with the figure. So please revise the text, caption and figure a bit to clarify the meaning of the circles and discussion.

17. Lines 312-314: I want to make sure I really understand figures 6,7,8, and the description of the figure is relatively terse in both text and captions. So I will state my guess about exactly what is being displayed, and please revise to clarify. Panel a is showing the POLDER annual mean for 2010. All fields are evaluated at identical points in space and time. Panel c is showing the mean of the emulator estimates for the annual averaged members of the ensemble. Panel d is showing the absolute value of 2* standard deviation of that ensemble of annual averages about the field displayed in panel c. Panel b is showing the difference between panels a and c. Panel e is showing panel b divided by panel d.

18. figure S2 caption. Is the minimum emission diameter for fossil fuels supposed to be 15nm? The caption indicates it is 25 nm.

19. Lines 450-476. I am struggling to extract clear messages, recommendations, and priorities from this section. You list again some of the deficiencies, and the parameters that influence them, but there is no recommendation about how to proceed in order to improve the situation, or a demonstration that a particular strategy could be used to improve the model fidelity. The section closes by indicating that more parameters could be examined, but there isn't really any indication that good use of the present ensemble of simulations could be used to improve the model or the model fidelity. Other than documenting the contribution of certain parameters to uncertainty, I am unsure of next steps. Could you discuss next steps?

20. Line 459: The sentence is unclear ("future work with subdivide")

---

## Author Comment (AC1)

**Response to Reviewers**

We appreciate the reviewers for their constructive comments and suggestions. Below we respond to each comment; reviewer comments are shown in black, our response is in *red italics*, and revised text is in blue. The line numbers we refer to are within the tracked changes document.

**Responses to Reviewer 1:**

**Specific comments:**

Lines 1-68: A first reading of the abstract and section 1 still left me wondering what the precise target was for the study. Could you state the goals a little more crisply? On line 6 you say your goal is to "address a gap" (my condensation on of your words). If I read your sentence carefully, I think you are saying that the gap is in quantifying the sources of uncertainty. So is the goal to identify some of the processes that contribute most to uncertainty in EC-H? And then compare those results to other models? You also eventually connect some of your results on parametric uncertainty with known structural deficiencies in the EC-H model. Maybe you could say something like these sentences explicitly, and provide hints on how (or whether) this information can be used to reduce uncertainty? I also disagree mildly with your characterization on of what structural uncertainty is, and how it contributes to model uncertainty (more below). I found the discussion of Regayre et al (2023) to have helped me understand what can and cannot be gleaned about structural and parametric uncertainty from PPE studies. Perhaps a little more of that kind of discussion could be inserted here also, since many authors are shared between that and this study.

We have modified the abstract and final paragraph of the introduction to better outline the goals and main findings of our study.

**Abstract modification:**

Interactions between aerosols, clouds, and radiation remain a major source of uncertainty in effective radiative forcing (ERF), limiting the accuracy of climate projections. This study aims to quantify uncertainties in aerosol—cloud and aerosol—radiation interactions using a perturbed parameter ensemble (PPE) of 221 simulations with the ECHAM6.3-HAM2.3 climate model, varying 23 aerosol-related parameters that control emissions, removal, chemistry, and microphysics.

The resulting global mean aerosol ERF is -1.24 W m-2 (5-95 percentile: -1.56 to -0.89 W m-2). Uncertainty in ERF is dominated by sulfate-related processes, biomass burning, aerosol size, and natural emissions. For aerosol-cloud interactions, dimethyl sulfide (DMS) and biomass burning emissions are key drivers, whereas sulfate chemistry and dry deposition exert the strongest influence on aerosol-radiation interactions. Despite structural differences across different models, the leading sources of ERF parametric uncertainty identified here are consistent with those found in other PPE studies, highlighting common sensitivities across climate models.

Comparison with POLDER-3/PARASOL satellite retrievals reveals persistent model biases in aerosol optical depth (AOD), Ångström exponent (AE), and single-scattering albedo (SSA), many of which fall within the parametric uncertainty range. Sulfate-related processes account for over 40% of AOD uncertainty, while AE and SSA are most sensitive to DMS, sea salt, and black carbon parameters. Correlation analysis between key parameters and observables indicates that several biases are tunable through physically consistent parameter adjustments. Our results highlight the need for combined efforts in parameter optimization and structural model development to improve confidence in aerosol-forcing estimates and future climate projections.

**Introductory modification:**

The primary aim of this study is to quantify parametric uncertainties in ERF and observable aerosol properties in ECHAM6-HAM, and to attribute the contributions of different aerosol and cloud parameters to these uncertainties. Based on this, we identify possible shared sources of uncertainty with other aerosol PPE studies.

2. Lines 10-11: I had a somewhat different interpretation of your results than expressed in this sentence. I think your sentence could be read to indicate that the leading causes of ERF uncertainty are associated with parameter uncertainties. I am not sure this is the case because some of your results indicate that structural deficiencies play a very large role in ERF uncertainty also, and you have by no means investigated many aspects of structural uncertainty. I personally believe structural uncertainty plays a very large role. I do think it is true that of the parametric uncertain es that you examined there were many common features with other PPE studies. Could you make your arguments more compelling or revise the sentence to agree more with my statement?

**This sentence has been modified to read:**

Despite structural differences across models, the leading causes of ERF parametric uncertainty identified here align with parametric uncertainties from other PPEs.

3. Lines 15-16: states "PPEs can reduce some structural model biases through parameter adjustments, but others persist." Maybe this topic deserves more discussion. Such adjustments might lead to a positive outcome, but isn't this fixing a problem for the wrong reason with the wrong solution? Presumably one doesn't like to correct a problem by adding an arbitrary correction of a (possibly) satisfactory parameterization to produce a reasonable result with a model containing a signifiant structural deficiency. It is unsatisfactory if the goal is to improve understanding and representation of the underlying physics, and it is at best an expedient kludge to avoid some other problem.

**This sentence has been removed and replaced with:**

Correlation analysis between key parameters and observables indicates that several biases are tunable through physically consistent parameter adjustments for bias reduction.

4. Lines 43-45: Identifying structural uncertainty with deficiencies in "coding" feels misleading. It could be that a process treatment represents one process very accurately, but entirely ignores another. Is this a coding problem? It is an "understanding problem" reflecting a lack of

understanding or some other choice by the team responsible for treating those processes. Can you replace "coded" with "represented", "treated", "formulated", etc?

**done**

5. Lines 54-55: The phrase "before observational constraints" doesn't deliver a clear message. Maybe "alongside observational uncertainties that constrain parameter choices", or some other phrasing.

**Sentence has been modified to:**

Causes of model uncertainty must be comprehensively quantified alongside observational uncertainties that constrain parameter choices to ensure their individual and combined effects on aerosol ERF are well understood (Yoshioka et al., 2019; Carslaw et al., 2013; Lee et al., 2013a, 2011; Regayre et al., 2015, 2014).

- 6. Line 65: It seems to me that the paper has two main goals:
  - a. Characterize the uncertain es for the EC-H model
  - b. Identify some of the shared parametric uncertainties with other models and PPE studies.

It might be worth adding a sentence or two here making these points before discussing what is going to be shown in various sections.

**We have updated this paragraph to better reflect these points:**

The primary aim of this study is to quantify parametric uncertainties in ERF and observable aerosol properties in ECHAM6-HAM, and to attribute the contributions of different aerosol and cloud parameters to these uncertainties. Based on this, we will identify possible shared sources of uncertainty with other aerosol PPE studies. In Section 2, we describe the ECHAM6.3-HAM2.3 climate model configuration used here, and the experimental setup of our perturbed parameter ensemble. Section 3.1 quantifies ERF uncertainties and attributes their respective causes, while Section 3.2 compares these PPE-based uncertainties against satellite observations to evaluate model performance and bias.

7. Lines 103-106: Are you partioning the ARI and ACI contributions to ERF using clear-sky and all sky fluxes, or some more sophisticated method (e.g., Ghan (2013), or an APRP method)? Please specify.

We have added a new Section in the supplementary materials (Section S1) in defining the ERF, ERFaci, ERFari calculations. This PPE did not output aerosol-free TOA diagnostics and therefore unable to perform the calculations from Ghan (2013). This has been fixed for future PPEs when constraining ERF. We have also added to main text text, including referencing to the calculation:

To calculate the total aerosol ERF and its components due to aerosol-cloud (ERFaci) and aerosol-radiation interactions (ERFari), we computed the radiative differences between these ensembles (2010 and 1850). We partition ERF into ARI and ACI by differencing all-sky and clear-sky top-of-atmosphere fluxes, such that ERFaci equals the PD–PI change in the shortwave cloud radiative effect and ERFari equals the PD–PI change in the clear-sky net flux. See Section S1 for more details on the calculation of ERF and the partitioning of ARI and ACI. This diagnostic approach provides a straightforward separation between cloud-mediated and clear-sky contributions. However, it may introduce biases relative to double-radiation-call or APRP decompositions (Taylor et al., 2007), which may be implemented for future work.

8. Line 179: perhaps insert "study" after the word "each" and change "sample" to "samples"?

Modified sentence to improve readability after implementing both reviewer comments:

Climate model PPE studies, (ECHAM6-HAM, CESM2-CAM6, HadGEM-UKCA; Yoshioka et al., 2019; Regayre et al., 2023; Eidhammer et al., 2024) and machine learning approaches (Albright et al., 2021; Smith et al., 2021) all sample different uncertainties based on their structural code base and the perturbations applied.

9. Line 189: typo "perturbationsc"

**fixed**

10. Line 203-205. These lines make explicit that the study is not addressing structural uncertainty. I believe that previous sections of the text were vague on this point, and I also think that the later discussion of the role of the CDNC minimum do deliver messages that are relevant to structural uncertainty.

**This sentence has now been removed.**

11. Lines 195-206: I am quite interested in this discussion of common features, conclusions, and differences between the EC-H simulations and analysis done here and studies with other models. I think the authors did a nice job of summarizing some of those differences, and I hope that there is more discussion like this in the summary sections of the study.

We have expanded the discussion of comparing differences across other PPEs in the discussion section outlined below:

When comparing a PPE across multiple models, some structural inadequacies may be overcome by re-tuning model parameters. Despite substantial structural differences between ECHAM6-HAM and HadGEM, the magnitude and dominant parametric sources of aerosol ERF uncertainty overlap in agreement across the ensembles. For example, natural aerosol emissions contribute significantly to both aerosol ERF and ERFaci uncertainty. Additionally, parameters that are globally important in aerosol ERF (EMI\_ANTH\_SO2, EMI\_DMS, and EMI\_CMR\_BB) are consistent across models. However, key differences are also highlighted due to structural deviations, such as ECHAM6-HAM producing a stronger (more negative) aerosol ERF over land, while most other models have a stronger aerosol ERF over marine regions over persistent stratocumulus clouds (Shindell et al., 2013; Regayre et al., 2018).

12. Lines 216-220. Are you saying that the biases in marine stratocumulus in ECHAM6 are structural, unlike those in HADGEM, UKESM, and the models analyzed by Shindell? I am unclear about the messages in these lines.

besides the parametric uncertainties, structural uncertainties also play a role in marine stratocumulus regions. We suggest that the large ERF bias in ECHAM6-HAM over the North Pacific Ocean is not only from parametric contributions, but structural errors also play large a role.

**We have modified this section to now read:**

The North Pacific Ocean exhibits the strongest regional mean ERF among ocean basins, at -5.8 W m-2, accompanied by a significant parametric uncertainty of 0.58 W m-2 (with a 5-95 percentile range of 147%). Structural cloud biases in ECHAM6-HAM contribute to a stronger marine ACI forcing (the North Pacific of around -4 W m-2) than Regayre et al. (2018); Smith et al. (2020) and Shindell et al. (2013b), with the strongest forcings over the regions of persistent stratocumulus clouds (Neubauer et al., 2019).

13. Lines 220-223. Do you think that the smaller uncertainties you identified in the marine tropics are just due to the crudity of the convective parameterizations and their ACI treatments in all global models, or do you think that those representations are actually robust and we really understand and are representing those processes accurately?

The smaller uncertainties in the marine tropics could be partly attributed to either (a) the crude convective parametric uncertainty which hides much of the ERF uncertainty, or (b) the difference PI to PD change is not very large in this region, so doesn't show a large standard deviation across ensemble members. Perhaps including parameters that are sensitive to marine convection zones may increase the ERF uncertainty across these regions.

We have added "parametric" to highlight these uncertainties are attributed to parameter perturbation rather than the inclusion of structural components too.

**We have also added to the section below:**

High-latitude marine regions demonstrate considerable parametric uncertainty in the ERF due to aerosol-cloud interactions, while tropical marine regions show comparatively smaller parametric uncertainties (Figure 2d). The smaller uncertainties in the marine tropics could be partly attributed to either (a) the convective parameterization, which may hide part of the ERF uncertainty (Neubauer et al., 2019), or (b) the relatively small PI-PD aerosol change in this region

14. Line 233-305: I like the discussion in sections 3.1.1 and 3.1.2 a lot. Would it make sense to change the title of section 3.1.2 to men on "CDNC minimum as an example of Structural Uncertainty" rather than to list the sec on with the specific topic of minimum CDNC? That way you have a section dealing with parametric uncertainty and another dealing with structural uncertainty (with a big focus on CDNC lower bounds). As you point out in the first sentence or two of 3.1.2 there are other possible contributions. The lower bound on CDNC just provides a convenient and very easy example to demonstrate the existence of structural uncertainty. It might

be useful to list some of the possible explanations for the need for such a limiter (CDNC also depends on estimates of subgridscale vertical velocity and internal circulations within the clouds driving aerosol activation, and unresolved or under-resolved exchanges between cloud free and cloudy air masses at cloud top. Other candidates that occur to me that are driven by radiative cooling, evaporation and drop sedimention. These are processes that occur to me, but you may have a better list).

We have changed the title of subsection 3.1.2 accordingly. In addition, we have expanded the discussion to highlight that the need for a relatively high CDNC lower bound as it reflects unresolved structural aspects of the model. The revised text now reads as follows.

Uncertainties shown in Figures 2c and 2d include only parametric uncertainties and exclude structural model uncertainties. Structural limitations, such as missing or oversimplified aerosolcloud processes, can introduce persistent biases and significantly alter simulated ERF (Regayre et al., 2023). One example is the minimum cloud droplet number concentration (CDNCmin= 40 cm-3) used in ECHAM6-HAM to maintain a realistic top-of-atmosphere energy balance (Lohmann and Neubauer, 2018; Neubauer et al., 2019). The need for such a relatively large CDNCmin suggests that it compensates for structural deficiencies or missing processes in the model. Several mechanisms may explain the use of this large value for CDNCmin: 1) smaller CDNC concentrations were observed mostly in much smaller pockets of cloud or regions than the grid box (Terai et al., 2014; Wood et al., 2018). 2) The treatment of secondary organic aerosol is simplistic and may underestimate the organic aerosol concentration in addition to a lack of representation of nitrate aerosol (Zhang et al., 2012). 3) The use of CDNCmin can cause a biased representation of liquid water pathway, stronger cloud phase feedback, and the entrainment rate for shallow convection (Lohmann and Neubauer, 2018; Neubauer et al., 2019). 4) There may be structural uncertainties in ice crystal nucleation in cirrus clouds or the stratocumulus cloud cover that can be tuned through CDNCmin (Neubauer et al., 2019). Together, these processes demonstrate how structural simplifications in cloud microphysics and subgrid dynamics can influence ERF estimates beyond the range captured by parameter perturbations alone (Neubauer et al., 2019).

15. Lines 301-305: I don't really understand the philosophy behind the proposed PPE study described on the lines of this paragraph. Are you just proposing an alternate kludge (to the CDNC lower bound), to avoid dealing with lack of understanding of the important processes? I could imagine employing a Machine Learning or AI method to connect CDNC, aerosol proper es, and meteorology/cloud proper es, but those words and methodologies are not mentioned on these lines, and it is hard for me to envision how one would use a PPE to improve structural deficiencies, as hinted at on these lines. Can you be more explicit about what you are proposing, or drop the paragraph?

We propose a follow-on study that extends the PPE with an additional focus on the cloud activation scheme without using a CDNCmin parameter and use a lower limit value.

We have slightly modified this paragraph, as below:

For a future PPE, including a perturbation of parameters in the activation and cloud activation schemes, we propose to impose no (or a very small) limit on CDNCmin and constrain the PPE with observations of both CDNC and aerosol parameters (including proxy for CCN), to achieve observationally constrained aerosol ERF that may overcome the need to adjust for structural model features. We expect this future work will constrain natural emission parameters towards higher values, leading in-turn to a smaller PI-PD increase in CDNC and weaker aerosol ERF, in line with Regayre et al. (2023).

16. Lines 306-320: I am assuming the black circles on each line (box/whisker plot) of figure 4 represent the value for a quantity and region for a particular member of the PPE. If I am correct, could you please state that in the caption? And if that supposition is correct, why do all the ensemble member values for AOD, AE and SSA cluster at the extreme ranges of the statistics, rather than frequently falling within the 25 and 75 percentile ranges? Since the circles never seem to fall within the interquartile range it suggests my stated supposition is incorrect, or there is a problem with the figure. So please revise the text, caption and figure a bit to clarify the meaning of the circles and discussion.

The black circles represent the value for ensemble members outside of the statistical range represented by the box and whisker box  $(Q1 - 1.5 \times IQR)$  and  $Q3 + 1.5 \times IQR)$ . We have changed the figure caption to clarify:

Regional direct comparison between the mean (a) Aerosol Optical Depth, (b) Ångström, and (c) Single Scattering Albedo observation from POLDER (red dot) with the box and whisker distribution from the PPE. The rectangular box represents the interquartile range (IQR; Q1 25th to 75th percentile of the PPE distribution), with the red vertical line representing the median of the PPE. The whiskers are defined by Q1 – 1.5 × IQR and Q3 + 1.5 × IQR. Ensemble members outside this defined statistical range (statistical outliers) are represented by black circles. Uncertainties from POLDER over land for (a) AOD is  $\pm$  0.04, (b) AE is  $\pm$  0.4, and (c) SSA is  $\pm$  0.03. The uncertainties from POLDER over the ocean for (a) AOD is  $\pm$  0.03, (b) AE is  $\pm$  0.25, and (c) SSA is  $\pm$  0.03. POLDER values taken over Antarctica, Antarctic Ocean, Arctic Land, Arctic Ocean are not considered in this work.

17. Lines 312-314: I want to make sure I really understand figures 6,7,8, and the description of the figure is relatively terse in both text and captions. So I will state my guess about exactly what is being displayed, and please revise to clarify. Panel a is showing the POLDER annual mean for 2010. All fields are evaluated at identical points in space and time. Panel c is showing the mean of the emulator estimates for the annual averaged members of the ensemble. Panel d is showing the absolute value of 2\* standard deviation of that ensemble of annual averages about the field displayed in panel c. Panel b is showing the difference between panels a and c. Panel e is showing panel b divided by panel d.

This description is correct apart from the relative uncertainty. This was calculated as absolute uncertainty divided by the mean multiplied by 100. We have updated the captions to better describe the figure. We have modified the figure caption for Figure 6, Figure 7, and Figure 8.

We also added a sentence at the beginning of the section:

The emulated PPE is co-located, in time and space, with the PARASOL satellite measurements using 3-hourly means to mitigate spatial and temporal sampling bias (Schutgens et al., 2016)

Caption: Figure 6. Global annual mean Aerosol Optical Depth (AOD 550 nm) for (a) POLDER retrievals, (c) PPE (co-located in space/time at 3-hourly intervals), and (b) the difference between the co-located PPE and POLDER (panel c minus panel a). The spatial magnitude of parametric uncertainty is shown as (d) PPE absolute uncertainty (2σ across the ensemble members), and (e) PPE relative uncertainty (absolute uncertainty (panel d) divided by PPE mean (panel c) multiplied by 100). Stippling indicates where the difference is not statistically significant at the 95% level of confidence student's t-test.

18. figure S2 cap on. Is the minimum emission diameter for fossil fuels supposed to be 15nm? The cap on indicates it is 25 nm.

This was a typo. The caption has been updated to reflect the correct minimum emission diameter for fossil fuels of 15 nm

30 nm \* 0.5 = 15 nm.

19. Lines 450-476. I am struggling to extract clear messages, recommendations, and priorities from this section. You list again some of the deficiencies, and the parameters that influence them, but there is no recommendation about how to proceed in order to improve the situation, or a demonstration that a particular strategy could be used to improve the model fidelity. The section closes by indicating that more parameters could be examined, but there isn't really any indica on that good use of the present ensemble of simulations could be used to improve the model or the model fidelity. Other than documenting the contribution of certain parameters to uncertainty, I am unsure of next steps. Could you discuss next steps?

We have added a new figure (Figure 9) to better discuss the next steps and overall parameter correlation to AOD, SSA, and AE.

Figure 9. Regional and global correlation coefficients (r) between each perturbed parameter and the emulated diagnostics (a) AOD, (b) SSA, and (c) AE for present-day conditions. Positive correlations (red) indicate that increasing the parameter value enhances the diagnostic, whereas negative correlations (blue) indicate a reduction. Rows show the parameters, and columns are the regions.

Figure 9 illustrates the linear correlations between each perturbed parameter and the simulated AOD, SSA, and AE across each region and globally. Positive correlations indicate that increasing the parameter value increases the diagnostic, while negative correlations imply an opposite effect. Through Figure 9, future modeling studies can apply tuning exercises based on their region of interest and tuning parameter to enhance or reduce their desired diagnostic.

Globally, AOD (Figure 9a) exhibits the strongest positive relationships with emission scaling factors, particularly natural emissions (DMS and sea salt), indicating that higher emissions generally increase aerosol loading. SSA (Figure 9b), in contrast, shows negative correlations with increasing biomass burning emission and black carbon refractive-index scaling, confirming that stronger BC absorption reduces single-scattering albedo. AE (Figure 9c) is most strongly influenced by dust and sea- salt emission scaling factors, with negative correlations suggesting that increases in these sources reduce AE through the introduction of coarser particles. These trends imply that, while total aerosol mass may be broadly consistent with observations, the effective size representation of coarse particles (dust and sea salt) likely contributes to the residual bias. In ECHAM6-HAM, the aerosol size for these species is prescribed within mode widths and is independent of the emitted mass (Tegen et al., 2019).

Therefore, adjustments to the emission size distribution rather than the total emission flux may reduce the AE and AOD bias. Regionally, the strongest sensitivities occur over source regions such as Africa and Asia for dust and biomass burning, and over the Southern Ocean for sea salt and DMS. These patterns highlight that addressing aerosol size representation, particularly for

natural coarse modes, is a crucial step in reducing uncertainty in modeled aerosol-radiation and aerosol-cloud interactions.

20. Line 459: The sentence is unclear ("future work with subdivide")

**Modified to**

Additionally, future work will focus more on sulfate-related parameters to better capture and constrain sulfates global uncertainty sources.

**Responses to Reviewer 2:**

This work presents a novel PPE design using the ECHAM6.3-HAM2.3 designed to characterize uncertainty in aerosol ERF. The parameters chosen in this work focus heavily on aerosol processes, including aerosol emissions, refractive indices, hygroscopicity, deposition, and nucleation/secondary aerosol processes. Across their PPE, they generate emulators for ERF, AOD, SSA, and AE and conduct regional analysis to identify the largest regional sources of parametric uncertainty for each diagnostic. They further validate their PPE against recent aerosol satellite observations to identify areas of bias in their simulations and point out areas of structural and parametric uncertainty.

Overall, I found the paper to be well written and easy to follow. The results were interesting and linked well to broader context of PPE work done in other models such as UKESM1, HadGEM, and CESM. I think they did a good job of setting the stage for future work looking at model improvements and development in ECHAM6.3-HAM2.3, but at times I found the results to be somewhat vague in the context of model improvement. Below, I offer some minor revisions and some clarification, but I think the paper should be published promptly with some minor changes.

**Major comments:**

Section 2.2.1: I would like to see an additional paragraph in this section that explains significance of the parameters in the context of the model. It doesn't have to be parameter by parameter, but it would be nice if the different categories of parameter were acknowledged. While the role of some is straightforward (emission, deposition, refractive index), I find it surprising that kappaSO2 and kappaSS don't contribute to cloud activation and would like to know briefly why that is and what role these values play for aerosol. The significance of PH\_PERT also eludes me in this context (controlling aqueous reaction rate?) so it would be nice to hear why these was chosen and what they represent.

A new paragraph has been added to section 2.2.1 to provide additional information on the parameters chosen for perturbation. In addition, more information is provided on the role of water uptake in aerosol. The new paragraph reads:

The parameters that are chosen, outlined in Table 1, span several categories that capture different aspects of aerosol-climate interactions in ECHAM6-HAM. Emission parameters (e.g., fossil fuel, biomass burning, biofuel, DMS, sea salt, dust) control the magnitude and composition of primary and precursor aerosol sources, directly affecting the aerosol burden. Size-related emission parameters (e.g., geometric mean diameters for different sources) determine the initial size distribution of emitted particles. Dry and wet deposition rates represent removal processes. Optical parameters, such as the imaginary refractive indices of black carbon and dust, affect the absorption and scattering of radiation. Chemical and microphysical processes are represented by parameters including the sulfate reaction rates, nucleation rates, and cloud water pH, which modifies the acidity of cloud droplets and thus aqueous-phase reaction rates, controlling sulfate production. Finally, hygroscopicity parameters (KAPPA\_SS, KAPPA\_SO4) describe aerosol water uptake used for aerosol optical calculations via kappa-Köhler theory. In ECHAM6-HAM, cloud droplet activation is computed by the Abdul-Razzak and Ghan (2000) scheme, which uses its own internal equations to determine when aerosols activate into droplets. Therefore, changing KAPPA values

mainly affects optical properties (like scattering and water uptake), rather than directly influencing the cloud activation process itself.

There is a lack of emission-size separation in this work for some of the more uncertain parameters such as sea salt and dust. The authors comment that this may be a direction for future work, but I think it would be interesting for the authors to comment on what the impacts of isolating size modifications for dust and sea salt could be given that mass emissions are changing both your aerosol size and your burden. In some cases, it seems the solution to fixing size biases may be to emit more mass, but the authors don't comment on whether the issue may lie in the size parameterization of these aerosol. Please see specific references in the minor comments below.

We have addressed parts of the size separation in later comments. Our future work will involve separating emission size accumulation and coarse mode emissions of sea salt.

Section 3.2.4: Most of the paper gives overall parameter uncertainty/sensitivity in different regions but little has been said about the sign of the parametric changes and how this relates to diagnostics in the model. From a model development side, I'd imagine it would be good to say what direction a parameter should be changed in order to address a given bias. Having this in the following section would be nice for the key uncertain parameters (e.g., 'future work would address increasing emissions of x and decreasing emissions of y to address a given bias'). Could you generate a simple linear correlation that targets the key parameters identified in your AOD, AE, and SSA uncertainty quantification (sections 3.2.1-3.2.3) to understand what direction they need to be tuned?

We have created a new figure showing the linear correlations for parameter perturbations and model output across different regions. We have also added new discussion points based on this plot at the end of section 3.2.4.

Figure 9. Regional and global correlation coefficients (r) between each perturbed parameter and the emulated diagnostics (a) AOD, (b) SSA, and (c) AE for present-day conditions. Positive correlations (red) indicate that increasing the parameter value enhances the diagnostic, whereas negative correlations (blue) indicate a reduction. Rows show the parameters, and columns are the regions.

Figure 9 illustrates the linear correlations between each perturbed parameter and the simulated AOD, SSA, and AE across each region and globally. Positive correlations indicate that increasing the parameter value increases the diagnostic, while negative correlations imply an opposite effect. Through Figure 9, future modeling studies can apply tuning exercises based on their region of interest and tuning parameter to enhance or reduce their desired diagnostic.

Globally, AOD (Figure 9a) exhibits the strongest positive relationships with emission scaling factors, particularly natural emissions (DMS and sea salt), indicating that higher emissions generally increase aerosol loading. SSA (Figure 9b), in contrast, shows strong negative correlations with increasing biomass burning emission and black carbon refractive-index scaling, confirming that stronger BC absorption reduces single-scattering albedo. AE (Figure 9c) is most strongly influenced by dust and sea- salt emission scaling factors, with negative correlations suggesting that increases in these sources reduce AE through the introduction of coarser particles. These trends imply that, while total aerosol mass may be broadly consistent with observations, the effective size representation of coarse particles (dust and sea salt) likely contributes to the residual bias. In ECHAM6-HAM, the aerosol size for these species is prescribed within mode widths that are independent of the emitted mass (Tegen et al., 2019).

Therefore, adjustments to the emission size distribution rather than the total emission flux may reduce the AE and AOD bias. Regionally, the strongest sensitivities occur over source regions such as Africa and Asia for dust and biomass burning, and over the Southern Ocean for sea salt

and DMS. These patterns highlight that addressing aerosol size representation, particularly for natural coarse modes, is a crucial step in reducing uncertainty in modeled aerosol-radiation and aerosol-cloud interactions.

Additionally, we have included a new sentence in the discussion/conclusion section to reflect this new result:

Therefore, we suggest that mitigating aerosol size bias in ECHAM6-HAM, particularly for natural coarse modes, is critical in reducing uncertainty in aerosol-radiation and aerosol-cloud interactions.

**Minor comments:**

Line 119: How many members have CDNCmin perturbed? Are these also randomly sampled, and are these members the only ones you use to train your GP emulator? I may be misunderstanding how CDNCmin is accounted for here and would appreciate more clarification.

Our training simulations (221 ensemble members) all include CDNCmin scale factors, which are uniformly sampled between 1 to 40 cm-3. After training the emulator, we only use the CDNCmin parameter values set to the default value of  $40 \text{cm}^{-3}$  for the main analysis of the manuscript. Only where stated otherwise in Section 3.1.2 is CDNCmin also included in the perturbation where we uniformly varied the perturbing parameter values.

We have re-phrased this paragraph to make it more clear of our procedure involving CDNCmin:

Across all 221 training simulation members, CDNCmin is also uniformly perturbed between 1 to 40 cm-3. After training the GP emulator, we set the CDNCmin parameter to the fixed sampling value of 40 cm-3, the default value in ECHAM6-HAM recommended by Neubauer et al. (2019), to quantify aerosol ERF uncertainty and the relative importance of the 23 parameters in Table 1 as causes of model uncertainty. In some instances, where explicitly stated (in Section 3.1.2), we also perturb CDNCmin, the minimum threshold for model cloud droplet concentrations, across 3 million model emulator-derived variants to highlight some of the structural and parametric uncertainties associated with clouds in ECHAM6-HAM. So, in Section 3.1.2 CDNCmin can be considered our 24th parameter, though we do not include it in Table 1 because its treatment is distinct from other parameters.

Table 1: If KAPPA\_SO4 and KAPPA\_SS are not used for cloud droplet activation, do these values only govern aerosol size? As a follow on, does ECHAM6-HAM account for aerosol hygroscopicity when parameterizing cloud droplet activation from aerosol? Upon second reading I'm wondering if ECHAM6-HAM uses two different hygroscopicities for aerosol and for cloud activation? Related to major comment regarding additional parameter explanation.

As detailed above, more details on the role from kappa SS and SO4 has been included within section 2.2.1 in the manuscript:

Finally, hygroscopicity parameters (KAPPA\_SS, KAPPA\_SO4) describe aerosol water uptake used for aerosol optical calculations via kappa-Köhler theory. In ECHAM6-HAM, cloud droplet activation is computed by the Abdul-Razzak and Ghan (2000) scheme, which uses its own internal

equations to determine when aerosols activate into droplets. therefore, changing KAPPA values mainly affects optical properties (like scattering and water uptake), rather than directly influencing the cloud activation process itself.

Line 178: In addition to references to previous PPE studies, please include the model acronyms associated with each study.

**done**

Line 209: Fig. 2b for the Europe ERF uncertainty? I would expect it to be higher than 0.41 when averaged over your Europe region based on this figure but I could be wrong. Please double check and be more clear in your figure references here.

This was a good catch. This lower reported value was in fact for the North Atlantic Ocean, and the updated value should be 0.68 W m-2. It should be noted that the uncertainty figure represents uncertainty as 2 \* std, and so this may present Europe with a larger standard deviation on the map than presented in the text. All other values were double checked and remain valid. To reflect this updated uncertainty value, we have also removed the point of "relatively low uncertainty", as the updated value reflects a reasonably high uncertainty for ERF.

Line 250: Please include reference to figure in this paper, as well as the paper you are referring to for HadGEM (Regayre et al., 2018, I'm assuming).

**Fixed**

Line 288: Regarding the comment on CDNCmin compensation for structural error: how do you rule out parametric uncertainty in this (i.e., from interactions of the other parameter choices)?

Sentence has been modified to make it clearer that you can not fully rule out parameteric uncertainty, as previously suggested:

The fact that such a large value of CDNCmin is needed suggests it is compensating for structural error and parameter choices in the model, either in the activation scheme or in terms of missing aerosol species or processes.

Line 311: When referring to Fig. 5, please also mention that it will be discussed more in sections 3.2.1-3.2.3.

**Sentence now reads:**

For example, Figure 5 shows the high contribution of natural aerosol emissions to global uncertainties of AOD, AE, and SSA, which will be discussed further in Sections 3.2.1 - 3.2.3.

Line 315: If this is true, what observational means are being shown in the Arctic and Antarctic regions in figure 4? Are you just showing available POLDER retrievals within your Arctic and Antarctic regions? Please clarify here.

Model bias is high over the Arctic and Antarctica, so we deemed it important to present the PPE results over this region. However, POLDER retrievals are biased over these regions and comparison to the PPE is not reliable. Therefore, we have excluded the POLDER result for these regions in Figure 4, as shown below:

We have also added a sentence in the figure caption to reflect this:

POLDER values taken over Antarctica, Antarctic Ocean, Arctic Land, Arctic Ocean are not considered in this work.

Figure 5: Check colors in these uncertainty plots and double check that they are showing the correct parametric value. Panel b shows two boxes for EMI\_BF (light green). The top one may be intended to be EMI\_FF?

This is a good catch. We have ensured Figure 3 and Figure 5 are using identical colors and markers for each parameter

Line 323-324: AOD underestimation appears to be related in large part to biomass burning regions. Coupled with the low uncertainty in many of these regions, is it possible that structural issues hinder representation of BB in ECHAM6-HAM and/or could this be due to too limited of range in some of the parameters chosen (BB emissions, BC refractive index)?

A key issue regarding biomass burning regions is the emission inventory used. The emission inventory used here was based on the default settings, the CMIP6 biomass burning inventory, making it a structural issue. For our future PPE on constraints, this issue can be mitigated using upto-date emissions from GFAS.

Line 327-328: This seems to indicate that there is very high parametric sensitivity in these regions in ECHAM6-HAM. Does this suggest that parameter choice in these regions is most important for constraining the PPE, or that there may be structural issues related to what I would assume are heavily anthropogenically influenced and dusty regions?

This analysis does not identify the origin of the model bias, but rather how the parametric uncertainties compare to model bias. This sentence suggests that the parametric uncertainties presented by the PPE has a larger magnitude than the model bias (structural or parametric in origin) in many regions. Therefore, it may be possible to constrain the some of the statistically significant model bias shown in Figure 6b through parametric tuning. We have added to this sentence as shown below:

Regions with high absolute AOD uncertainty correspond to areas with significant differences between the model and observational data, as shown in Figure 6d. Some regions, such as central and northern Africa, and Southeast Asia, have a larger parametric uncertainty (Figure 6d) than model bias (caused by combined structural and parametric uncertainties). This magnitude difference suggests it may be possible to constrain some of the model bias shown in Figure 6b through parametric tuning.

Line 337-338: Could this also be addressed by changing emission parameter ranges? Or is this deemed to put parameter ranges outside of what is physically sound?

Indeed, this is possible, but the extent of perturbing dust emissions would be outside a plausible range of emissions. Therefore, an alternative solution to mitigate the dust bias over Africa would be to increase more dust emission in the coarse mode and reduce the accumulation mode emissions. This is also highlighted in Figure 9, suggesting dust emissions over Africa need to increase in large modes, which would subsequently increase AOD.

Line 348: Consider rewording sentence, changing '...AOD bias are shown...' to '...AOD bias and are shown...'

**done**

Line 368: Figure reference should be Figure 6e, not Figure 6d

**fixed**

Line 379-380: Does this suggest that ari should not be addressed in isolation from aci?

This was not the intended outcome of our sentence. Therefore, we have removed the part of the sentence that gives this confusion, which now reads:

biomass burning-related parameters cause substantial perturbation to the ERF uncertainty (Figure 3), but not so for AOD uncertainty (Figure 5). Thus, observational constraints to match present-day AOD will not guarantee constraints on all parameters relevant to ERFaci.

Line 388-389: The trends in AE suggest that dust and sea salt aerosol are too small. There is high sensitivity to emissions, but emissions don't differentiate between mass and size impacts. Can you comment on how size is treated in ECHAM6-HAM? Is there any evidence to suggest that emissions may be appropriate, but the size treatment of dust and sea salt may be the issue?

Those trends in AE for dust and sea salt are consistent with the literature. Prior ECHAM6-HAM aerosol evaluations have identified a sea salt and dust mode issue, with Tegen et al. (2019) showing too high emissions in the accumulation mode and a lack in the coarse mode. Although the issue may not stem from the extent of emissions, as is the case for AOD, which has a negative bias over the Southern Ocean (Figure 6b), where sea salt typically dominates (e.g., Revell et al., 2021). However, a positive tropical AOD bias suggests the sea salt emissions may be too high (Tegen et al. 2019). The extent of appropriateness regarding sea salt emissions would require further evaluations with profile measurements, but we are confident in concluding that sea salt has too high accumulation mode emissions, and not enough coarse mode in its current configuration.

Evidence suggests that dust emissions may not be fully representative in the model, as models fail to simulate sporadic dust events. Dust emissions in ECHAM6-HAM are lower (1100 Mt yr-1) than the AeroCom average (1800 Mt yr-1), with a dust burden also slightly lower in ECHAM6-HAM (17 Tg) than in AeroCom (19.2 Tg) from Tegan et al. (2019).

Using Figure 9, we can see how much of an extent increasing sea salt and dust emissions reduce AE. We have discussed this further in Section 3.2.4. To work on this in the future, we are perturbing both the accumulation and coarse mode in sea salt emission to constrain these emissions. Additionally, we have added a new perturbation to the dust emission parameterization with a goal of mitigating emission magnitude and size uncertainties.

Much of this is now discussed in the new figure and discussion.

Line 410: What could the implications be for modifications to aerosol size in isolation of mass since this seems like a key issue in this area as well?

We have modified this sentence as there was some confusion on this, to suggest that increasing aerosol size in the Southern Ocean region would influence AE and AOD, as suggested by Figure 9.

Line 415: Does this mean that increasing emissions would accentuate the bias by increasing size? Are emissions the issue or is the size parameterization the issue? I'd be interested in some discussion on this topic.

We have added a 'See Section 3.2.4' in this sentence, as dust emissions are discussed in my detail in reference to Figure 9.

Line 416-418: Does this mean that increasing emissions would increase the bias or the other way around? I'm a little confused in this section on the sign of your parameter influence on the AE, AOD, and SSA diagnostics.

Yes, increasing the emissions would increase bias, as shown by Figure 9c, but this may possibly be compensated through changing the dust refractive index. Dust emissions are further discussed in Section 3.2.4, based on the new Figure. Additionally, the POLDER bias over this dust region plays an important in identifying model bias here, and newer satellites (e.g. PACE) will provide better measurements for AE over the Sahara region.

Line 439: Size and mixing state may also be playing a role. Can you comment on these parameters in ECHAM6-HAM and how they may be contributing to this bias? Does ECHAM6-HAM treat aerosol as volume mixed? That could also be overestimating absorption. Brown et al., 2021 (https://doi.org/10.1038/s41467-020-20482-9) showed an underestimation in SSA over biomass burning regions in this same model.

Based on Figure 5c and Figure 9a, biomass burning size (EMI\_CMR\_BB) does not have a large a role on the magnitude of uncertainty and bias, but the bias likely stems mostly from the imaginary refractive indices of BC. Figure 9a does suggest larger EMI\_CMR\_BB can cause a lower AOD, but the extent of which is not substantial. For our next PPE, we have extended the range of perturbing biomass burning emitted size beyond what is used in this study. We also note from Figure 9b that biomass burning emitted size does not contribute significantly to SSA, but rather the refractive index of BC and emission of BB has a large role in affecting SSA (strongly inversely correlated). Finally, ECHAM6-HAM uses volume-weighted averaging with each aerosol mode internally mixed.

We have added some text in Section 4.2.3 to further discuss the role of biomass burning to SSA:

SSA (Figure 9b), in contrast, shows negative correlations with increasing biomass burning emission and black carbon refractive-index scaling, confirming that stronger BC absorption reduces single-scattering albedo.

Line 445-447: Similar to above comment that size and mixing state may be playing a role.

Reference to Figure 9b has been added to this sentence, and further elaboration is made in Section 4.2.3, as shown in the above reply.

Line 455: As a very minor comment, consider rewording 'smaller size distributions' to 'smaller aerosol'. I say this given that a distribution can vary in number, width, and size which makes smaller seem somewhat arbitrary to me.

**done**

Line 473: What does '(a lack of)' refer to? Please reword to be more clear.

**Fixed to read:**

The cause of this mismatch is likely from structural uncertainty stemming from issues in coarse mode aerosol emissions for larger aerosol sizes, as shown by Tegan et al. (2019)

Line 473: As per previous comment, consider rewording 'larger aerosol size distributions' to 'larger aerosol sizes' or 'larger aerosol diameters'.

done

---

## Author Response (AR2)

We appreciate the reviewer for their secondary suggestions. Below we respond to each comment; reviewer comments are shown in black, our response is in *red italics*, and revised text is in blue.

Reviewer #1

The authors have attempted to address all of my comments. Thank you. I offer a few more minor suggestions where I think the presentation could be improved a bit more, to make points more easily understood.

Minor comments:

line 4: Insert "parametric" before uncertainties.

*done*

line 12: Change "Despite structural uncertainties" to "Structural Uncertainty is difficult to characterize, and this study focuses primarily upon an evaluation of parametric uncertainty"

*done*

line 20: Change "are tunable" to "may be reduced by tuning"

*done*

line 50: Insert the phrase "(parametric uncertainty)" after the word "parameterizations"

*done*

line 146: Insert "To provide some insight into the model sensitivity to structural uncertainties," before the sentence, and delete the phrase "as causes of model uncertainty" at the end of the sentence.

*done*

lines 251-254: I am still unhappy with the phrasing of this sentence. Here is a suggested rough rewrite of the sentence that I think would clarify the messaging (and feel free to fix it up if I have missed some of the points). "The remaining ERF_aci biases in ECHAM6-HAM in that region that are not explained by parametric uncertainties (and hence are attributed to structural deficiencies) have an amplitude of around 4W/m2. This bias is somewhat different from the models analyzed in Shindell et al(), Regayre et al(), etc who noted large residual structural biases in persistent subtropical stratocumulus regions.

*This sentence has now been modified to read:*

The remaining ERFaci biases in ECHAM6-HAM in that region, which are not explained by parametric uncertainties (and hence are attributed to structural deficiencies), have an amplitude of approximately 4 W m−2 over the regions of persistent stratocumulus clouds (Neubauer et al., 2019). This bias differs somewhat from the models analyzed in Regayre et al. (2018); Smith et al. (2020) and Shindell et al. (2013).